# Can Extracts from the Leaves and Fruits of the *Cotoneaster* Species Be Considered Promising Anti-Acne Agents?

**DOI:** 10.3390/molecules27092907

**Published:** 2022-05-02

**Authors:** Barbara Krzemińska, Michał P. Dybowski, Katarzyna Klimek, Rafał Typek, Małgorzata Miazga-Karska, Grażyna Ginalska, Katarzyna Dos Santos Szewczyk

**Affiliations:** 1Department of Pharmaceutical Botany, Medical University of Lublin, 20-093 Lublin, Poland; barbara.krzeminska@umlub.pl; 2Department of Chromatography, Faculty of Chemistry, Institute of Chemical Sciences, Maria Curie Sklodowska University in Lublin, 20-031 Lublin, Poland; michal.dybowski@poczta.umcs.lublin.pl (M.P.D.); rafal.typek@poczta.umcs.lublin.pl (R.T.); 3Department of Biochemistry and Biotechnology, Medical University of Lublin, 20-093 Lublin, Poland; katarzyna.klimek@umlub.pl (K.K.); malgorzata.miazga-karska@umlub.pl (M.M.-K.); grazyna.ginalska@umlub.pl (G.G.)

**Keywords:** Cotoneaster, Rosaceae, skin diseases, anti-acne, LC-MS, antioxidant, anti-inflammatory activity

## Abstract

This study aimed to evaluate the phenolic profile and biological activity of the extracts from the leaves and fruits of *Cotoneaster nebrodensis* and *Cotoneaster roseus*. Considering that miscellaneous species of *Cotoneaster* are thought to be healing in traditional Asian medicine, we assumed that this uninvestigated species may reveal significant therapeutic properties. Here, we report the simultaneous assessment of chemical composition as well as biological activities (antioxidant, anti-inflammatory, antibacterial, and cytotoxic properties) of tested species. Complementary LC-MS analysis revealed that polyphenols (especially flavonoids and proanthocyanidins) are the overriding phytochemicals with the greatest significance in tested biological activities. In vitro chemical tests considering biological activities revealed that obtained results showed different values depending on concentration, extraction solvent as well as phenolic content. Biological assays demonstrated that the investigated extracts possessed antibacterial properties and were not cytotoxic toward normal skin fibroblasts. Given the obtained results, we concluded that knowledge of the chemical composition and biological activities of investigated species are important to achieve a better understanding of the utilization of these plants in traditional medicine and be useful for further research in their application to treat various diseases, such as skin disorders.

## 1. Introduction

Oxidative stress plays a pivotal role in morbidity among people with multifarious diseases. There is significant research involving the influence of oxidative stress on the impairment of cells, tissues, and even the whole body. Moreover, many reports suggest that free radicals are involved in the triggering of various disorders, such as arteriosclerosis, Alzheimer’s, and Parkinson’s diseases, as well as many types of cancers [1]. The available data suggest that the diminution in oxidative stress induced by reactive oxygen species (ROS) and reactive nitrogen species (RNS) is beneficial to human health. Therefore, many attempts have been made to discover effective antioxidants with an acceptable safety profile. The fact that synthetic antioxidants can reveal remarkable side effects may lead to pose a question about the appropriateness of using them in large quantities, especially during the long term. Latest studies reported by Xu and co-authors [2] confirmed the importance of the control of the amounts of synthetic phenolic antioxidants, such as butylated hydroxyanisole (BHA), butylated hydroxytoluene (BHT), tertiary butylhydroquinone (TBHQ), and propyl gallate (PC), which are consumed despite that their excessive quantities can damage DNA as well as lead to apoptosis and carcinogenicity [2].

To date, there is an extensive body of literature involving radical scavenging by polyphenols (more than 700 papers in a timeframe from 1995 to 2009) [1]. For instance, Sakar et al. reported that antioxidants such as flavonoids revealed various biological activities such as anti-inflammatory, antiviral, antibacterial, vasodilatory, and anticancer properties [3]. According to Haritwal and co-authors [4], mRNAs of several antioxidant enzymes (catalase, glutathione transferase, glutathione peroxidase, and superoxide dismutase) that occur in cells can be upregulated by polyphenols contained in several plants (e.g., *Phyllanthus amarus*), which, as a consequence, led to the neutralization of oxidative stress. Taking into consideration that Sakar and co-authors [3] highlighted the importance and the possibility of application of plant agents in various disorders, the use of the extracts abundant in phenolic compounds emerges as a viable alternative for synthetic antioxidants. Another noteworthy observation is that several antioxidants, such as flavonoids (especially apigenin and quercetin), have been suggested as a complementary therapy for COVID-19, thanks to their multidimensional actions [5].

Therefore, antioxidants of plant origin, in the most respects, are becoming more approved than synthetic ones, as they seem to possess lower toxicity and higher efficacy. Thus, there is crucial importance to provide new sources of natural compounds with significant antioxidant profiles [4]. Moreover, these compounds play an important role in the treatment of skin disorders such as acne, a chronic disease concerning the pilosebaceous unit that reveals as different kinds of imperfections, e.g., nodules, cysts, papules, pustules, comedones, and sinuses [6,7]. According to Soleymani et al. [8], approximately 50 million people in the United States suffer from acne. Additionally, the cost of treatment of acne is estimated at nearly $3 billion per year. Despite the popular belief that acne is a condition primarily present during adolescence, the prevalence of acne among adult women ranges between 12 and 54% [7,8]. 

Even though the etiopathogenesis of acne is distinctly complex and not fully explained, the four substantial factors affecting this lesion can be distinguished, namely: abnormal follicular hyperkeratinization, excess sebum, inflammatory mechanisms, as well as the proliferation and colonization of *Propionibacterium acnes* (recently called *Cutibacterium acnes*) in the pilosebaceous unit [8,9]. 

In recent times, scientists focused on the discovery of antioxidant substances from plants that work effectively against various diseases without significant side effects [10]. External use of medications in the treatment of acne may cause skin irritation. In addition, the use of antibiotics may contribute to a significant increase in antibiotic resistance. Therefore, active substances of plant origin with proven anti-inflammatory, antioxidant, sebum regulation, and antimicrobial properties are of great importance [11].

The data described above confirm the importance of conducting investigations on effective phytochemicals in acne treatment. Furthermore, in the light of current issues, namely the increasing antibiotic resistance [12] and intensification in the occurrence of acne due to long-time mask-wearing during the coronavirus pandemic [13,14], the search for new herbal substances with anti-acne properties that can be used for a long period of time with minimal risk of side effects appears to be justified.

Among various plants that arouse the interest of scientists, the *Cotoneaster* genus seems to be a promising target for future investigation for a few reasons. The *Cotoneaster* species belong to the Rosaceae family, whose representatives are rich in active ingredients (especially phenolic compounds), both qualitatively and quantitatively. Moreover, the plants that belong to *Cotoneaster* are increasingly recognized as promising antioxidant agents, abundant in phenolic compounds. The results obtained by Kicel et al. [15] suggested that among the twelve *Cotoneaster* species cultivated in Poland, *C. integerrimus* possesses the highest level of flavonoids. Moreover, the highest total content of polyphenols (especially mono-, di-, and trimeric flavan-3-ol derivatives) was observed in *C. bullatus* as well as in *C. zabelii*. Furthermore, according to Kicel and co-authors [16], diethyl-ether fraction of *C. zabelii* and ethyl acetate fraction of *C. bullatus* and *C. integerrimus* revealed significant free radical scavenging ability at a concentration of 50 µg/mL. Furthermore, our previous research [17] provided a description of two cultivated *Cotoneaster* species as promising candidates for potential use in skin disorders, such as acne.

Moreover, *Cotoneaster* leaves and fruits are widely used in traditional medicine as cooling, astringent, and expectorant agents [18]. These plants are also valued as antiviral, anticancer [19], cardiotonic, and diuretic drugs [20,21]. Some *Cotoneasters* are also used in the treatment of abdominal pain, eye diseases, piles, leucoderma, itches, fevers, cuts, and wounds [18,22]. Additionally, they have numerous applications in nasal hemorrhage, excessive menstruation, and diabetes mellitus [19,23]. Additionally, the application of *Cotoneaster* species in traditional Asian medicine, including the treatment of skin disorders, is of great significance [16,24].

*C. roseus* and *C. nebrodensis* have not yet been studied. Thus, gaps in knowledge involving them in the available literature can be observed. Based on the above, the aim of the present study was to fill the knowledge gap about features of *C. roseus* and *C. nebrodensis* by the establishment of the chemical composition as well as biological activities (antioxidant, antibacterial, enzyme inhibitory, cytotoxic properties) of untested extracts from the leaves of these species. We assumed that this plant material may be a promising source of bioactive phytochemicals (polyphenols), which can support the treatment of acne. The knowledge of the occurrence of free radicals scavengers in investigated species of *Cotoneaster* will have high importance for a greater understanding of the possibility of applying these plants in traditional Asian medicine, as well as will be useful for employment them in dietary supplements and natural-based products, which focus on chronic diseases connected with oxidative stress.

## 2. Results and Discussion

### 2.1. Phytochemical Analysis

Total phenolic content (TPC) in *Cotoneaster nebrodensis* and *C. roseus* extracts was determined using Folin–Ciocalteu reagent, and the results were estimated as gallic acid equivalents (GAE) per g of dry extract (DE) (Table 1). Our results showed that the fruits (CRMF) and the leaves (CRML) of *C. roseus* had the highest phenolic content (132.45 ± 0.21 and 118.43 ± 0.41 mg GAE/g DE, respectively). In our previous study, we received higher amounts of phenolic compounds in the methanol–acetone–water (3:1:1, *v*/*v*) extracts of the leaves of *C. hissaricus* (296.13 ± 1.52 mg GAE/g DE) and *C. hsingshangensis* (193.84 ± 1.14 mg GAE/g DE), also cultivated in Poland [17]. Kicel et al. [15] found that the total phenolic content in the 70% aqueous methanolic extracts obtained from the leaves of various *Cotoneaster* species cultivated in Poland ranged from 51.7 (*C. tomentosus*) to 154.3 mg GAE/g DW (*C. bullatus*). These authors [16] also demonstrated lower TPC levels in the fruits (from 26 to 43.5 mg GAE/g DW) of various *Cotoneaster* species.

The total flavonoid content in the leaves and fruits of *C. nebrodensis* and *C. roseus* was determined using the previously described colorimetric method [25]. The data were expressed as the quercetin equivalents (QE) per g of dry extract (DE). The results presented in Table 1 show that the highest content of total flavonoids was in both extracts obtained from the leaves of *C. roseus* (51.60 ± 0.71 for CRML and 37.58 ± 0.70 mg QE/g DE for CREL). These reached data were higher than those determined in our previous study for extracts of the leaves of *C. hissaricus* and *C. hsingshangensis* [17].

Mahmutović-Dizdarević et al. [26] found that the methanolic extracts from the leaves of *C. tomentosus* contained 18.17 ± 0.30 mg QE/g of dry weight (dw.) total flavonoid level, followed by *C. integerrimus*—16.42 ± 0.35 mg QE/g dw. and *C. horizontalis*—10.55 ± 0.51 mg QE/g dw., and their results were comparable with those obtained for the leaves of *Cotoneaster nebrodensis* (17.21 ± 0.29 mg QE/g DE). The results obtained by these authors for fruits were lower (2.76–9.38 mg QE/g dw.) compared to the data noted in our study (1.90–25.40 mg QE/g DE). A comparable level of flavonoids was found in the methanolic extract of leaves of *C. wilsonii* Nakai (36.46 ± 1.89 mg QE/g dw), while in the stems and fruits, the content of flavonoids was lower (6.09 ± 0.71 and 0.23 ± 0.20 mg QE/g dw, respectively) [27]. In our study, the lowest amount of total flavonoids was found in ethanol extracts of fruits and leaves of *C. nebrodensis* (1.90 ± 0.32 and 3.53 ± 0.09 mg QE/g DE, respectively).

The total phenolic acid content (TPAC) in *Cotoneaster* extracts was presented in Table 1. Higher content of phenolic acids was noted for the methanol–acetone–water extracts obtained from the fruits and leaves of *C. roseus* (59.79 ± 0.42 and 57.41 ± 0.39 mg CAE/g DE, respectively). These results are close to the data obtained in our previous study for the crude extract of leaves of *C. hsingshangensis* (61.27 ± 0.93 mg CAE/g DE) and higher than those obtained for leaves of *C. hissaricus* (33.80 ± 1.03 mg CAE/g DE) [17].

In the next step of our study, the chemical composition of the extracts obtained from the leaves and fruits of *C. nebrodensis* and *C. roseus* was investigated using the LC-MS method. Appendix A shows the identified compounds, including their molecular formula, theoretical and experimental molecular mass, both errors in ppm and mDa, and the fragments.

The results of the LC-MS analysis obtained for extracts from the leaves and fruits of *C. nebrodensis* and *C. roseus* are listed in Table 2 and Table 3. The amounts of the identified compounds were carried out using the calibration curves obtained for the standards. In the case of the quantitative analysis of compounds, in which appropriate standards were not available, the calibration curves for substances of the similar structure were used.

Among the identified flavonoids in the leaf extracts, quercetin derivatives were the most abundant in both species. In the methanol–acetone–water and 60% ethanol extracts from the leaves of *C. nebrodensis* quercitrin (1734.0 ± 76.2 and 1274.5 ± 21.5 μg/g DE, respectively), astragalin (1468.7 ± 49.1 and 1170.0 ± 42.1 μg/g DE, respectively), and isoquercitrin (1064.5 ± 43.1 and 852.5 ± 42.5 μg/g DE, respectively) were found in the largest amount. However, much higher amounts of flavonoids were noticed in both extracts from the leaves of *C. roseus*. In the CRML and CREL extracts, isoquercitrin (6520.1 ± 234.0 and 4991.3 ± 174.8 μg/g DE, respectively), hyperoside (4141.0 ± 152.1 and 3086.5 ± 126.0 μg/g DE, respectively), rutin (3443.0 ± 143.3 and 1095.1 ± 64.2 μg/g DE, respectively), and vitexin 2″-*O*-arabinoside (2867.4 ± 117.0 and 1869.5 ± 75.3 μg/g DE, respectively) were observed in the greatest amount. Hyperoside, isoquercitrin, and rutin were previously identified as dominant compounds in the other *Cotoneaster* species [15,16,17,19,20,28,29,30,31,32,33]. Vitexin 2″-*O*-arabinoside, which was found in large quantities in *C. roseus*, was previously noticed only in our previous study in the leaves of *C. hissaricus* and *C. hsingshangensis* [17] and in the leaves of *C. thymaefolia* [29]. 

It is worth noting that despite their rarity in nature flavonoids, 5-methylgenistein-4′-*O*-glucoside and sissotrin were observed in both studied species. The first ones were previously observed only in the methanol–acetone–water extract from the leaves of *C. hissaricus* and *C. hsingshangensis* [17] and chloroform extract from the leafy twigs of *C. simonsii* [29]. Sissotrin was found in our previous study in the leaves of *C. hissaricus* and *C. hsingshangensis* [17], and also in the leaves of *C. mongolica* [32], flowers of *C. serotina* [33], and the flowers and fruits of *C. pannosa* [33]. Moreover, the occurrence of astragalin in the *Cotoneaster* genus is usually rare, and it was reported only in the methanol–acetone–water extract of *C. hissaricus* and *C. hsingshangensis* leaves [17] and the methanol extract of *C. mongolica* leaves [32].

Among the identified phenolic acids, chlorogenic acid was the most abundant in both species. A significant amounts of this acid were noticed in CRML (26,836.5 ± 987.0 μg/g DE), and CREL (21,822.0 ± 584.0 μg/g DE) extracts. Taking into consideration the data obtained by Kicel and co-authors, it can be concluded that chlorogenic acid occurs in various *Cotoneaster* species. The UHPLC-PDA-ESI-QTOF-MS analysis proved that the occurrence of this acid in 70% aqueous methanolic extracts from the leaves of *C. integerrimus, C. tomentosus, C. melanocarpus, C. lucidus, C. divaricatus, C. horizontalis, C. nanshan, C. hjelmqvistii, C. dielsianus, C. splenden, C. bullatus*, and *C. zabelii* [15]. Similarly, in our previous research [17], we found significant amounts of chlorogenic acid in the leaves of *C. hissaricus* and *C. hsingshangensis*. It is worth highlighting that chlorogenic acid possesses antibacterial activity against *S. dysenteria* and *S. pneumoniae*, and also has other biological activities such as hepatoprotective, cardioprotective, antiphlogistic, antimutagenic, antiviral, anti-inflammatory, anti-obesity, and anti-hypertension properties [34,35,36].

Apart from typical phenolic acids, cotonoate A, horizontoate A, and horizontoate C were observed in all extracts from the leaves of both species. The highest amounts of these compounds were noticed in the CRML extract (1034.0 ± 42.1, 1175.2 ± 49.3, and 1743.0 ± 63.6 μg/g DE, respectively). In our previous study, only cotonoate A was identified in quantifiable amounts (1564 ± 55 μg/g DE) in the leaves of *C. hissaricus.* Moreover, cotonoate A was isolated from the leafy twigs of the *C. racemiflora* [37], and horizontoate A and C was found in the methanolic extract from *C. horizontalis* [18].

Among the other polyphenolic compounds, some coumarins were also identified in the methanol–acetone–water extracts from the leaves of *C. roseus* and *C. nebrodensis,* wherein scopoletin was the most abundant in both species (8880.0 ± 331.2 and 4143.5 ± 152.2 μg/g DE, respectively). These amounts were lower than those found in our previous study for the leaves of *C. hissaricus* and *C. hsingshangensis* (12,219 ± 440 and 10,481 ± 371 μg/g DE, respectively) [17]. Scopoletin was also noticed previously in the leafy twigs of *C. racemiflora* [38].

In the methanol–acetone–water extract from the leaves of *C. roseus*, a significant amount of mannitol was also observed (1189.3 ± 41.0 μg/g DE). However, its amount was much lower than that found by us in the leaves of *C. hissaricus* (6834 ± 249 μg/g DE) and *C. hsingshangensis* (3104 ± 123 μg/g DE) [17]. From cyanogenic glycosides, prunasin was found in all extracts. The largest amount of this compound was noticed in the CRML extract (1145.7 ± 41.2 μg/g DE). The second identified cyanogenic glycoside, amygdalin was observed only in the methanol–acetone–water extract from the leaves of *C. roseus* (538.6 ± 23.0 μg/g DE). These compounds were previously identified in the leaves of *C. hissaricus* and *C. hsingshangensis* [17], leafy twigs of *C. horizontalis* [39], and the fruits and leaves of *C. congesta*, *C. praecox*, and *C. integerrimus* [40].

The qualitative and quantitative composition of the extracts obtained from the fruits of both studied species was different compared to the composition of the extracts from the leaves. The results of the analysis of the fruits of *C. nebrodensis* and *C. roseus* are presented in Table 3 and in Figure 1, Figure 2, Figure 3 and Figure 4.

In the case of the methanol–acetone–water extract from the fruits of *C. nebrodensis,* the highest compound concentration was observed for two flavonoids, i.e., (+)-catechin (8940.7 ± 377.2 μg/g DE) and naringin (8917.7 ± 409.3 μg/g DE). Three phenolic acids, i.e., 5-*O*-caffeoylquinic acid (chlorogenic acid), cinnamic acid, and 3-*O*-*p*-coumaroylquinic acid, were determined at a concentration of 5907.0 ± 294.2, 5411.6 ± 246.8, and 4268.3 ± 205.3 μg/g DE, respectively. Procyanidin C-1, caffeic acid, and biochanin (5,7-dihydroxy-4-methoxyisoflavone) were determined at the lowest concentration level—below 10 µg/mL. The total concentration of all identified polyphenolic compounds in the CNMF extract was about 40 mg/g. In the 60% ethanol extract from the fruits of *C. nebrodensis,* the highest compound concentration was determined for cinnamic acid (3532.8 ± 151.2 μg/g DE). A slightly lower analyte concentration was observed for 5-*O*-caffeoylquinic acid (chlorogenic acid) and (+)-catechin (1703.5 ± 80.9 and 1006.1 ± 42.1 μg/g DE, respectively). The lowest concentration (below 0.1 mg/g) was determined in the case of other polyphenols. The total concentration of all identified polyphenols in the 60% ethanol extract from the fruits of *C. nebrodensis* was 10 mg/g. In the methanol–acetone–water extract from the fruits of *C. roseus*, the highest analyte concentration was determined in the case of three compounds, i.e., cinnamic acid (2846.8 ± 120.4 μg/g DE) and its derivatives, 5-*O*-caffeoylquinic acid (chlorogenic acid; 24,124.0 ± 1153.1 μg/g DE), and 3-*O*-*p*-coumaroylquinic acid (27,744.0 ± 1315.1 μg/g DE). In turn, the concentration below 35 µg/g was determined for syringic acid, vanilic, and sinapic acid. The total concentration of all non-volatile identified compounds in the examined extract was slightly above 9 mg/g. Similarly to the methanol–acetone–water extract from the fruits of *C. roseus*, the highest concentration of the same compounds (cinnamic acid and its derivatives, 5-*O*-caffeoylquinic acid, chlorogenic acid, and 3-*O*-*p*-coumaroylquinic acid) was determined in CREF extract, and their amounts ranged 2324.7–18,178.4 μg/g DE. Comparable results were observed for compounds with a concentration below 30 µg/g; additionally to this group, gallic acid, and C-1 procyanidin can be included. The total concentration of all non-volatile compounds identified in this extract exceeded 9 mg/g.

Considering the methanol–acetone–water and ethanolic extracts from given species, it should be noted that in the case of both *C. nebrodensis* and *C. roseus*, a higher total concentration of polyphenolic compounds has been found in *C. roseus*. The methanol–acetone–water extract obtained from the leaves and fruits had more than two times higher concentrations of the polyphenols, and the ethanolic extract was above five times higher concentration of these analytes. The comparison of the obtained data for these two species can notice that the methanol–acetone–water was a more effective solvent for the isolation of polyphenolic compounds from *Cotoneaster* compared to 60% ethanol. The concentration of the mentioned compounds in the methanol–acetone–water extracts in relation to ethanol extracts was approximately two times higher.

Among the compounds identified in *C. nebrodensis* and *C. roseus*, chlorogenic acid [41], cinnamic acid [42], ferulic acid [43], caffeic acid [44], vitexin [45], and quercetin [46] have been reported to show anti-acne activity.

### 2.2. Antioxidant Activity

Antioxidants play a pivotal role in the prevention of the damaging effects of free radicals. A large number of synthetic antioxidants can be distinguished; however, in many cases they can induce harmful effects. For this reason, looking for natural antioxidants of plant origin is of great significance [47]. Thus, our study was designed to establish the antioxidant effect of fruits and leaves of *C. roseus* and *C. nebrodensis*.

Various in vitro tests based on SET-(single electron transfer) and HAT-type (hydrogen atom transfer) mechanisms are widely used for assessment of antioxidant properties of phenolic compounds. Nevertheless, because ROS showed miscellaneous mechanisms of action, none of these methods can be considered universal [15]. Thus, the diversification of several tests is required. Among the methods based on SET-(single electron transfer) mechanisms, FRAP (ferric ion reducing antioxidant parameter), CUPRAC (cupric ion reducing antioxidant capacity), ABTS as well as DPPH can be distinguished. Moreover, HAT-type (hydrogen atom transfer) mechanism provides carrying out different tests, namely: ORAC (oxygen radical absorbance capacity), TRAP (total redox antioxidant parameter), CBA (crocin bleaching assay), and LPIC (lipid peroxidation inhibition capacity) [48].

Therefore, chemical methods do not consider the physiological conditions of pH and temperature, as well as the metabolism of antioxidants in the body or their transport within cells; thus, the research in biological models of human blood plasma is also required. This knowledge gap is excepted to be filled and seems to be a future research direction.

ROS such as hydroxyl, superoxide, and nitrous oxide provoke irritation in the area of acne lesions [49]. The clinical study conducted by Sarici and co-authors [50] indicates that oxidative stress may play a pivotal role in the etiology of acne. Spectrophotometric measurement of selected oxidative stress parameters, namely superoxide dismutase (SOD), nitric oxide (NO), catalase (CAT), malondialdehyde (MDA), and xanthine oxidase (XO) in the venous blood of the patients was performed. The level of CAT and SOD was significantly lower in the patient group than in the control group; simultaneously, the level of MDA as well as XO were significantly higher in the group consisting of patients with acne vulgaris.

According to Melnik et al. [51], the reactive oxygen species (ROS) contribute to p53 response and oxidative stress activates Sestrin 1 and Sestrin 2, which consequently leads to inhibition of mTORC1.

The data obtained by Soleymani and co-authors [8] indicated that phenolic compounds possess the ability to diminish oxidative stress via various pathways. Considering the above, the inhibition of endothelial ROS (reactive oxygen species) level, the reduction of ROS/MAPK/NF-ĸB as well as PI3K/Akt (protein kinase B)/NF-ĸB pathways and down-regulation of the gene expression of MAPKs (P38, ERK (extracellular signal regulated kinases), JNK (c-Jun N-terminal kinase)) can be distinguished. Additionally, the suppression of Nrf2 (nuclear factor erythroid 2—related factor)/Keapl-mediated antioxidant pathway and regulation of NOX-4, MDA (malondialdehyde), SOD (superoxide dismutase), GP x, GSH (glutathione peroxidase), CAT (catalase) level have been established. Moreover, the adjustment of PI3K/Akt and MAPK (mitogen-activated protein kinase) signaling pathways, the reduction of STAT-1 activation, and Nrf2-mediated HO-1 induction are of particular significance [52,53].

In comparison with previous studies [17], the plants investigated in our scrutiny seem to be promising candidates for future research due to the fact that they possess adequate antioxidant activity.

Due to the significant content of polyphenols presented in the above research, we assumed that the studied species may show significant antioxidant activity. Therefore, we decided to test their antioxidant activity using chemical methods, such as DPPH, ABTS, and CHEL. The antioxidant activity was studied on the microplate scale in cell-free systems. The extracts from *C. nebrodensis* and *C. roseus* were evaluated in a concentration ranging from 20 to 250 μg/mL. The results of antioxidant tests are presented in Table 4. It was demonstrated that extracts from the fruits and leaves of both investigated species exhibited moderate scavenging capacity in a concentration-dependent manner. The highest DPPH scavenging activity was found for the methanol–acetone–water extract from the fruits of *C. roseus* (IC_50_ = 22.94 ± 0.20 μg/mL), followed by the methanol–acetone–water extract from the leaves of *C. roseus*, and the ethanol extract from the fruits of *C. roseus* (IC_50_ = 32.12 ± 0.19 and 35.49 ± 0.50 μg/mL, respectively). The weakest activity was noted for the 60% ethanol extract from the fruits (CNEF; IC_50_ = 125.63 ± 0.02 μg/mL) and the leaves of *C. nebrodensis* (CNEL; IC_50_ = 117.79 ± 0.02 μg/mL). For comparison, the radical scavenging activity of ascorbic acid (AA; IC_50_ = 4.29 ± 0.09 μg/mL), quercetin (IC_50_ = 2.38 ± 0.11 μg/mL), and Trolox (IC_50_ = 3.74 ± 0.15 μg/mL) were tested at the same conditions.

Considering the DPPH method, it has been proven that various *Cotoneaster* species possess significant antioxidant activity. The ethanolic extract from the leafy twigs of *C. horizontalis* demonstrated the ability to scavenge free radicals, with EC_50_ equal to 19.3 µg/mL [39]. Aqueous, methanolic and ethyl acetate extracts obtained from the leafy twigs of *C. nummularia* showed EC_50_ in the range of 0.097–0.252 mg/mL [21]. Research conducted by Kicel and co-authors [15] is of great importance because it revealed the antioxidant activities of the methanol–water (7:3) extracts obtained from the leaves of 12 species cultivated in Poland, such as *C. melanocarpus* (EC_50_ = 32.75 µg/mL), *C. integerrimus* (EC_50_ = 24.58 µg/mL), *C. tomentosus* (EC_50_ = 34.50 µg/mL), *C. lucidus* (EC_50_ = 25.35 µg/mL), *C. divaricatus* (EC_50_ = 18.45 µg/mL), *C. horizontalis* (EC_50_ = 23.02 µg/mL), *C. nanshan* (EC_50_ = 24.68 µg/mL), *C. hjelmqvistii* (EC_50_ = 21.04 µg/mL), *C. dielsianus* (EC_50_ = 29.49 µg/mL), *C. splendens* (EC_50_ = 22.56 µg/mL), *C. bullatus* (EC_50_ = 20.91 µg/mL), and *C. zabelli* (EC_50_ = 21.52 µg/mL).

Moreover, hexane and ethanolic extracts obtained from the leaves of *C. afghanicus* showed EC_50_ in the range of 57.4–64.7 µg/mL [54]. Another study revealed antioxidant activity of ethanol–water (7:3) as well as ethanol–hexane (55:45) extracts from the fruits of *C. pannosus* with EC_50_ ranging 47.3–54.9 µg/mL [20].

In our previous research [17], we established the IC_50_ values for the methanol–acetone–water extracts from the leaves of *C. hissaricus* and *C. hsingshangensis* and the results were as follows: IC_50_ = 21.73 ± 0.13 µg/mL for *C. hissaricus* and IC_50_ = 10.16 ± 0.02 µg/mL for *C. hsingshangensis*. Furthermore, ethyl acetate fractions were the most promising candidates for future research concerning antioxidant activity, as the IC_50_ value for *C. hissaricus* fraction was 5.40 ± 0.10 µg/mL, while the ethyl acetate fraction for *C. hsingshangensis* showed better activity with IC_50_ = 2.08 ± 0.03 µg/mL.

To the best of our knowledge, there are only a few studies on the antioxidant activity of *Cotoneaster* species established by the ABTS method; thus, our study was designed to assess ABTS for investigated species. As shown in Table 4, the ABTS^●+^ assay revealed that the methanol–acetone–water extract from the fruits of *C. roseus* (CRMF) possessed the strongest ability to scavenge free radicals (IC_50_ = 10.89 ± 0.11 μg/mL), followed by the methanol–acetone–water extract (CRML; IC_50_ = 21.04 ± 0.11 μg/mL), and the 60% ethanol extract (CREL; IC_50_ = 31.98 ± 0.17 μg/mL) from the leaves of *C. roseus*. The weakest activity was noted for the 60% ethanol extract from the fruits of *C. nebrodensis* (CNEF; IC_50_ = 121.06 ± 0.21 μg/mL).

Zengin and co-authors [21] established that the extracts obtained from the leafy twigs of *C. nummularia* revealed EC_50_ = 0.043 mg/mL, EC_50_ = 0.020 mg/mL and EC_50_ = 0.023 mg/mL for ethyl acetate, methanol, and water extract, respectively. Mahmutović-Dizdarević et al. [26] determined IC_50_ values of the methanolic extracts obtained from the leaves and barks of *C. horizontalis*, *C. integerrimus*, as well as *C. tomentosum*. The IC_50_ for *C. integerrimus* leaves was 0.20 ± 0.01, while for bark 0.73 ± 0.02 mg/mL. The IC_50_ for extract from the leaves of *C. tomentosus* was 0.12 ± 0.01 and for extract from bark was 0.87 ± 0.03 mg/mL. The IC_50_ for *C. horizontalis* revealed 0.38 ± 0.01 and 0.42 ± 0.01 for leaves and bark extracts, respectively.

According to the ABTS results demonstrated by Ali and co-authors [55], the methanol extract obtained from aerial parts of *C. microphyllus*, revealed IC_50_ = 92 µg/mL, while IC_50_ for its subfractions were as follows: ethyl acetate—178 µg/mL, chloroform—220 µg/mL, and n-hexane—880 µg/mL. At the same time, the methanol extracts obtained from roots of *C. microphyllus*, showed IC_50_ = 90 µg/mL and IC_50_ for its subfractions equal to 178 µg/mL for ethyl acetate, 220 µg/mL for chloroform, and 880 µg/mL for n-hexane subfraction.

In our previous study [17], methanol–acetone–water extracts obtained from the leaves of *C. hsingshangensis* as well as of *C. hissaricus* demonstrated IC_50_ values in ABTS test equal to 1.92 ± 0.13 µg/mL and 4.14 ± 0.10 µg/mL, respectively. Among various subfractions of these extracts, the most active in ABTS test were ethyl acetate fractions, with IC_50_ = 0.90 ± 0.02 µg/mL for *C. hissaricus* and 0.37 ± 0.01 µg/mL for *C. hsingshangensis*.

As shown in Table 4, the extracts from the fruits and leaves of both investigated *Cotoneaster* species possessed a weak capacity to interfere with the formation of iron and ferrozine complexes. The IC_50_ values of all extracts showed lower chelating activity than the positive control–Na_2_EDTA*2H_2_O (IC_50_ = 4.69 ± 0.17 µg/mL). Among the studied extracts, the best activity was noticed for the methanol–acetone–water extract from the fruits of *C. roseus* (CRMF; IC_50_ = 29.62 ± 0.23 µg/mL). As far as metal chelating assay is concerned, Zengin et al. [21] reported that aqueous, methanolic, and ethyl acetate extracts from the leafy twigs of *C. nummularia* revealed activity, with EC_50_ = 0.3–18.7 mg EDTAE/g. Scrutiny conducted by Ekin et al. during metal chelating assay [30] revealed that the ethanolic extract from the leaves of *C. nummularia* possesses an inhibition activity equal to 26.2% (2 mg/mL; the extract concentration). Moreover, the same study demonstrated that ethanolic extract from the leaves of *C. meyeri* and *C. morulus* showed inhibition properties close to 5.9% and 21.5% (2 mg/mL; the extract concentration), respectively.

According to the data obtained by Uysal et al. within CHEL test [56], aqueous and methanolic extracts from twigs and fruits of *C. integerrimus*, possessed EC_50_ = 1.47–6.24 mg/mL and EC_50_ = 2.14–6.14 mg/mL, respectively.

In our previous study [17], we attempted to estimate IC_50_ values for methanol–acetone–water extract from *C. hissaricus* and *C. hsingshangensis,* as well as for their various subfractions. Our scrutiny revealed significant metal chelating activity of subfractions of *C. hsingshangensis*, namely ethyl acetate fraction (IC_50_ = 0.50 ± 0.01 μg/mL) and butanol fraction (IC_50_ = 1.01 ± 0.01 μg/mL).

### 2.3. Enzyme Inhibitory Activity

An extensive body of literature discusses the issue of the role of inflammation in acne formation. Contrary to previous assumptions, the inflammatory process can be found at the very beginning of acne lesions development. Lymphocytic infiltrate including CD4+ T-cells as well as CD68+ macrophages leads to generation of microcomedones. Neutrophils take part in the inflammation at a later stage and contribute to the occurrence of pustules. The production of IL-1α, which is involved in IL-1α-induced hyperkeratinisation, can be observed within comedones. Moreover, cytokines and chemokines associated with the Th17 pathway, such as IL-1β, IL-6 and TGF-β, TNF-α, IL-8, CSF2, and CCL20 have been reported as agents involved in the development of inflammation [57].

Inhibition of the following inflammation pathways: IL-1β, IL-6, NF-ĸB, TNF-α (tumor necrosis factor), JAK2/STAT3, TRAF1/ASK1/JNK/NF-ĸB, NF-ĸB, AP-1 (activator protein), PI3K/Akt (protein kinase B)/NF-ĸB, TAK1—NF-ĸB, JAKs/STAT1 as well as NOX2/p47 signaling pathways is caused by phenolic compounds. Moreover, inhibition of metalloproteinases such as MMP 13, MMP-2 and MMP-9, as well as VCAM-1 and ICAM-1 is noteworthy [8].

In order to investigate the potential anti-inflammatory activity of the tested *Cotoneaster* extracts, inhibition of lipoxygenase (LOX), COX-1, COX-2 as well as hyaluronidase (HYAL) was evaluated (Table 5).

The presence of hyaluronic acid within the blood vessel walls is of great importance due to the fact that the permeability of capillary walls is directly proportional to hyaluronic acid content. This compound is deactivated by the enzyme hyaluronidase [58]. 

The scrutiny of the in vitro study conducted by Kuppusamy et al. [59] demonstrated that aglycons possessed stronger inhibition activity compared to glycosides. Therefore, quercetin inhibited hyaluronidase to a greater extent than rutin. Interestingly, the in vivo research did not confirm this thesis because both aglycons, as well as glycosides, revealed indistinguishable activity. Thus, the hydrolysis of glycosides in the body may play a pivotal role in this respect to hyaluronidase [58].

Taking into consideration that the chemical composition of *C. roseus* and *C. nebrodensis* showed the presence of flavone compounds and that flavone compounds reduced hyaluronidase activity [58], we decided to investigate hyaluronidase inhibitory activity of the examined samples.

In our study, the methanol–acetone–water extract from the fruits of *C. roseus* (CRMF) exhibited the best hyaluronidase inhibition activity (IC_50_ = 13.96 ± 0.11 µg/mL) compared to the other extracts (Table 5). Moderate activity was also noted for the methanol–acetone–water extract (CNMF; IC_50_ = 23.69 ± 0.19 μg/mL) and for the 60% ethanol extract (CNEF; IC_50_ = 24.07 ± 0.09 μg/mL) from the fruits of *C. nebrodensis*. However, the activity of these extracts was lower compared to positive standard—Indomethacin (IC_50_ = 7.23 ± 0.02 μg/mL).

Many authors noted that plants such as *Hedera helix*, *Aesculus hippocastanum*, *Ruscus aculeatus*, or *Glycyrrhiza glabra* possessed the capacity to provide flavonoid-rich extracts, abundant in triterpenic saponins and sapogenins with significant hyaluronidase inhibitory activity hyaluronidase [58,60,61,62].

In accordance with the available literature on *Cotoneaster*, it has been proven that some species seem to be promising pro-inflammatory agents, e.g., methanol–water extracts from the leaves of *C. zabelii* and *C. bullatus* showed IC_50_ =7.9–8.1 µg/mL for HYAL. Interestingly, for these species, the anti-hyaluronidase properties turned out to be higher or not statistically different than obtained for positive controls (IC_50_ = 8.6 µg/mL) [16]. Additionally, the methanol–water extracts obtained from the fruits of *C. lucidus* revealed strong anti-hyaluronidase properties (IC_50_ = 25.7 µg/mL) [16]. Methanol–water (7:3) extract from the bark of *C. integerrimus* showed high anti-hyaluronidase activity with IC_50_ = 8.6 µg/mL [28]. Moreover, in our previous research, we noticed that the ethyl acetate fraction of *C. hsingshangensis* exhibited high hyaluronidase inhibition activity (IC_50_ = 1.89 μg/mL). The IC_50_ values for the crude extracts of *C. hissaricus* (15.09 ± 0.61 μg/mL) and *C. hsingshangensis* (6.82 ± 0.15 μg/mL) were also lower than those obtained from the fruits and leaves of *C. roseus* and *C. nebrodensis* [17].

PRAR ligands bind PPARα receptors occurring in sebocytes (within microsomes, mitochondria, and peroxisomes) and provoke lipogenesis. The look for active molecules, which possess anti-lipoxygenase properties, appears to be of particular importance, because 5-lipoxygenase inhibitors decrease lipogenesis and thus reduce acne [63].

The results of the inhibition of lipoxygenase by extracts from *C. roseus* and *C. nebrodensis* are presented in Table 5. The methanol–acetone–water extract from the fruits of *C. roseus* (CRMF) showed a considerable ability to inhibit lipoxygenase activity (IC_50_ = 74.62 ± 0.33 μg/mL), while the 60% ethanol extract from the fruits (CNEF; 577.90 ± 0.10 μg/mL) and leaves (CNEL; 324.95 ± 0.15 μg/mL) of *C. nebrodensis* showed the lowest activity. It is worth noting that CRMF extract exhibited higher inhibition activity compared to indomethacin (IND)—a positive standard (IC_50_ = 81.35 ± 0.23 μg/mL).

Kicel and co-authors [16] reported that extracts from the leaves of *C. zabelii* and *C. bullatus* appeared to be noteworthy anti-inflammatory factors as they revealed IC_50_ = 217.8–185.8 µg/mL for LOX. Furthermore, the data obtained by these authors [16] indicated that fruits of *C. hjelmqvistii* (IC_50_ = 290.0 µg/mL) and *C. zabelli* (375.9 µg/mL) showed strong anti-lipoxygenase properties. Methanol–water (7:3) extract from bark of *C. integerrimus* showed high anti-lipoxygenase activity with IC_50_ = 169.0 µg/mL [28]. The results obtained by Krzemińska et al. [17] showed that diethyl ether fraction and ethyl acetate fraction of *C. hsingshangensis* had a high ability to inhibit lipoxygenase activity (IC_50_ = 4.15 and 5.72 μg/mL, respectively). 

COX-1 and COX-2 (cyclooxygenases) are answerable for the transformation of arachidonic acid into several pro-inflammatory mediators, such as prostaglandin H2 (PGH2). Thus, these enzymes play a pivotal role in the inflammatory process [64].

It has been established that various plants from the Rosaceae family are capable of inhibiting the activity of COX-1 as well as COX-2 enzyme. According to Dos Santos Szewczyk et al. [65], various extracts and subfractions (60% methanol, diethyl ether, ethyl acetate, and *n*-butanol) obtained from aerial parts and roots of *Alchemilla acutiloba*, possessed significant cyclooxygenase inhibitory activity. Particularly noteworthy is the fact that butanol fraction of the methanolic extract from roots of *A. acutiloba* at the concentration of 100 µg/mL inhibited 83.14 ± 1.08% of COX-1 activity and simultaneously declined 95.10 ± 1.81% of COX-2 activity.

To the best of our knowledge, just a few reports regarding cyclooxygenase-1 (COX-1) and cyclooxygenase-2 (COX-2) inhibitory activity of *Cotoneaster* species have been conducted. Thus, our study was designed to fill this gap in knowledge. To determine the potential anti-inflammatory activity of *C. nebrodensis* and *C. roseus*, we also evaluated the ability of the extracts to inhibit the conversion of arachidonic acid to PGH_2_ by ovine COX-1 and human recombinant COX-2 using a COX inhibitor screening assay kit (Cayman Chemical, MI, USA) (Table 5). The most active extracts against COX-1 were CNEL (IC_50_ = 14.31 ± 0.38 µg/mL) and CREL (IC_50_ = 19.15 ± 0.45 µg/mL), while the weakest was CREF (IC_50_ = 102.88 ± 0.15 µg/mL). In the case of COX-2, CNMF (IC_50_ = 10.44 ± 0.06 µg/mL) and CREL (IC_50_ = 16.00 ± 0.09 µg/mL) were the most active extracts followed by CNEL and CNEF (IC_50_ = 23.68 ± 0.27 and 29.81 ± 0.01 µg/mL, respectively).

According to our previous research [17], we investigated extracts as well as subfractions obtained from the leaves of *C. hissaricus* and *C. hsingshangensis,* and we proved that they showed significant activity against COX-1 and COX-2. The most active against COX-1 were the diethyl ether fraction of *C. hsingshangensis* (IC_50_ = 6.39 μg/mL) and the ethyl acetate fraction of *C. hsingshangensis* (IC_50_ = 9.54 μg/mL). Simultaneously, methanol–acetone–water extract of *C. hsingshangensis* (IC_50_ = 9.21 ± 0.09 μg/mL) as well as the diethyl ether fraction of *C. hsingshangensis* (IC_50_ = 5.09 ± 0.06 μg/mL) revealed the best inhibitory activity towards COX-2 among all investigated extracts and subfractions of both species. Moreover, Zengi and co-authors [22] attempted to investigate the influence of the methanol and aqueous extracts obtained from the fruits and twigs of *C. integerrimus* on the functioning of several inflammatory cytokines, such as PGE2 (the main product of COX-2), 5-HT, and TNFα.

### 2.4. Antibacterial Activity

Various mechanisms are involved in the formation of acne with the participation of *C. acnes* (previously known as *P. acnes*). The expression of toll-like receptors (TLRs) and protease-activated receptors (PARs) can be distinguished. Moreover, *C. acnes* contributes to upregulation of tumour necrosis factor (TNF) and the secretion of matrix metalloproteinases (MMPs), TNF, interferon γ (INF-γ), and interleukins (IL-8, IL-12, IL-1) by keratinocytes [66,67,68].

Walsh and co-authors [69] indicated that the issue of *C. acnes* resistance to popular antibiotics using both oral and topical, such as tetracyclines, macrolides, or lincosamines appear to be a significant worldwide emergence [69]. As noted by McLaughlin et al. [57], resistant strains belong predominantly to the type IA1 and IC clade; however, strains from the type IA2, IB, and II phylogroups also reveal resistance but to a lesser extent.

Secondary metabolites derived from various plants, including phenolic compounds, distinctly indicate their anti-acne potential. Artonin E (flavonoid) and pyranocycloartobiloxanthone A, obtained from *Artocarpus elasticus*, revealed significant antimicrobial properties (MIC = 2.0 μg/mL) but lower than standard antibiotic—Clindamycin (MIC = 0.03 μg/mL) [70]. A combination of tetracycline and ellagic acid (ETC; 250 μg/mL + 0.312 μg/mL) efficiently inhibited biofilm formation by *C. acnes* (approximately 80–91%) and provided significant improvement in susceptibility to antibiotics [71]. The results obtained by Batubara et al. [72] clearly indicated that flavonoid fustin possessed an efficient ability to inhibit the *C. acnes* lipase activity. Fustin was obtained from *Intsia palembanica,* and the 2,3-dimercapto-1-propanol tributyrate (BALB) method was employed to establish the lipase activity properties.

Based on the above, the search for new active ingredients with antibacterial activities is of particular importance.

Both the leaves and fruits of *C. nebrodensis* and *C. roseus* species were analyzed for antibacterial properties, including their anti-acne properties. Methanol–acetone–water and 60% ethanol extracts of these plant materials were used for the tests. The diffusion tests, in which 100 µg of samples were applied, demonstrated that the methanol–acetone–water extract from the fruit of *C. nebrodensis* (CNMF) and ethanol extract (CNEF) was the most active against acne microaerobic strains, with the zones of growth inhibition in the range of 17–15 mm and 14–10 mm, respectively (Figure 5). The leaves of *C. roseus* also had moderate activity against these bacteria, with growth inhibition zones equal to 15–11 mm and 14–12 mm for CRML and CREL extracts, respectively. Moreover, both extracts of *C. roseus* fruits (CRMF, CREF) exhibited a slight anti-acne activity against *Cutibacterium* spp., with the growth inhibition zone in the range of 14–9 mm. On the other hand, all tested fractions from the leaves of *C. nebrodensis* (CNML and CNEL) showed no significant activity. The inhibition activity of the tested extracts from the fruits and leaves against Gram-positive *S. aureus* and *S. epidermidis* was as follows: the most active was methanol–acetone–water extract from the fruits of *C. nebrodensis* (19–15 mm), then CREL (18–12 mm), CNEF (12–10 mm), CRMF and CRML (10–8 mm). The other samples had no significant effect on aerobic Gram-positive bacteria. The tested extracts had a narrow spectrum of activity directed only at Gram-positive strains; none of them inhibited the growth of *E. coli*.

The next stage of the microbiological experiments involved the assessment of the minimal inhibitory concentration (MIC) of the extracts that showed antibacterial activity in the previous experiment. The results presented in Table 6 confirmed that the methanol–acetone–water extract from the fruit of *C. nebrodensis* (CNMF) had the strongest properties against all tested Gram-positive strains, with the MIC value ranging 250–500 µg/mL. The CRML extract also achieved favourable MIC values (1000–4000 µg/mL). The remaining samples had less beneficial MIC values, namely CREL, CNEF, and CRMF extracts possessed MIC values in the range from 1000 up to more than 4000 µg/mL. Gallic acid was used as a positive control in antimicrobial experiments. It exhibited a wide spectrum of antimicrobial activity, involving not only Gram-positive aerobic and anaerobic strains, but also Gram-negative *E. coli* (MIC 2000 µg/mL). Nevertheless, it is worth underlining that CNMF extract exhibited better MIC values for all Gram-positive strains compared to values determined for gallic acid (Table 6).

As reported by Sati and co-authors [73], the diffusion-disk method revealed promising antibacterial properties of the ethanolic extract of *C. acuminatus* roots at the concentration of 100 µg/mL against *Bacillus subtilis*, *Bacillus pumilus*, *Staphylococcus aureus*, *Microccocus glutamicus*, *Pseudomonas aeruginosa*, *Proteus vulgaris*, and *Escherichia coli* with growth inhibition zones ranged 10–18 mm. Inhibition zone of *C. nummularioides* methanol extract was 12 mm for *B. cereus* at the concentration of 400 mg/mL [74].

Studies conducted by Bukhari and co-authors [54] revealed that the fixed oils, as well as essential oils obtained from *C. afghanicus* at the concentration of 1 mg/mL, inhibited the growth of *Bacillus subtilis* (zone of inhibition equal to 7 ± 0.08 and 5 ± 0.07 mm, respectively). Moreover, ethanol extract of *C. afghanicus* (1 mg/mL) possessed antibacterial activity towards *B. subtilis* (zone of inhibition 6 ± 0.09 mm) and *Escherichia coli* (zone of inhibition 3 ± 0.04 mm).

Methanolic extracts from *C. nummularioides*, *C. draba*, and *C. dactylon* were assessed with regard to antibacterial properties. MIC values for *C. nummularioides* showed the best results and ranged from 3.125 to 66.667 mg/mL [74]. According to Uysal and co-authors [56], the fruit methanolic extract of *C. integrifolius* revealed significant antimicrobial activity with the MIC value ranged from 0.195 to 6.25 mg/mL. This sample turned out to be especially active toward MRSA strains of *S. aureus,* with MIC values ranging from 0.195 to 0.391 mg/mL.

### 2.5. Cytotoxic Activity

Among the extracts obtained from the leaves, CNML the most potently inhibited viability of normal fibroblasts (Figure 6), with CC_50_ value close to 125 μg/mL (Table 7). In turn, other extracts obtained from the leaves were non-cytotoxic towards BJ cells (CC_50_~1000 μg/mL). In the case of the extracts obtained from the fruits, it was found that cell viability after treatment with CNMF and CRMF extracts (CC_50_~300-350 μg/mL) was lower than after incubation with CNEF and CREF (CC_50_ > 1000 μg/mL). Moreover, the calculated values of therapeutic indexes (TIs) indicated which extracts were the safest in vitro. The TI value equal to 1 (TI = 1) indicates that the tested extract reduces the viability of bacteria and mammalian cells in the same manner, while TI above 1 (TI > 1) shows that the extract was more active towards bacteria than eukaryotic cells [75]. Thus, the CRML extract inhibited the growth of *C. acnes* PCM 2400 more potently than it reduced the viability of fibroblasts (TI~2). In turn, the CNMF extract was more active towards *E. epidermidis* ATCC 12228, *C. acnes* PCM 2334, and *C. acnes* PCM 2400 than towards BJ cells (TI~1.2).

### 2.6. Hierarchical Cluster Analysis of the Phytochemical and Activity Data of C. nebrodensis and C. roseus

A hierarchical cluster analysis (HCA) was performed for all data obtained as a result of performed experiments. The Ward’s method and Euclidean squared distance was employed.

Taking into consideration the diverse levels of phytochemicals content as well as antioxidant and enzyme inhibitory activities of all tested samples, three distinct clusters (CLF1-CLF-3) can be distinguished (Figure 7).

In the CLF-1, two samples of the methanolic extract of the leaves and fruits of *C. roseus* (CRML, CRMF) were associated with the highest content of phenolic content and high antioxidant activity (DPPH, ABTS, CHEL) as well as LPO and HYAL activities (Table 8, Table 9 and Table 10).

The cluster 2 (CLF-2) encompasses four samples (CNML, CREL, CNMF, CREF), which is the ethanolic extract of *C. roseus* leaves and fruits, as well as the methanol–acetone–water extract of *C. nebrodensis* leaves and fruits. CLF-2 shows lower values of TPC, TPAC and TFC as well as higher values of DPPH, ABTS, CHEL, LPO, HYAL tests than CLF-1 (Table 8, Table 9 and Table 10).

In the CLF-3 cluster with two samples of ethanolic extract from the leaves and fruits of *C. nebrodensis* (CNEL, CNEF), the lowest values of TPC, TPAC, TFC, and the highest values of DPPH, ABTS, CHEL, LPO, HYAL were observed (Table 8, Table 9 and Table 10).

CLF-1 revealed the best profile of phenolic constituents and the most promising biological activities.

The COX1 values were lower within CLF-1 and CLF-3 as well as CLF-2 showed higher values. Taking into account COX2, the highest values were observed within CLF-1, lower in CLF-2 and the lowest in CLF-3.

Additionally, for a clearer visualization of the results, biological activities of tested extracts have been presented as box-whiskers plots (Figure 8).

## 3. Materials and Methods

### 3.1. Chemicals and Reagents

2,2-diphenyl-1-picrylhydrazyl radical (DPPH^•^), 2,2′-azino-bis-(3-ethyl-benzothiazole-6-sulfonic acid) (ABTS^●+^), indomethacin, ascorbic acid, hyaluronidase from bovine tests, hyaluronic acid sodium salt from rooster comb, Folin–Ciocalteu reagent, ethylene-diaminetetraacetic acid, disodium dihydrate (Na_2_EDTA*2H_2_O), Trolox (±)-6-hydroxy-2,5,7,8-tetramethylchromane-2-carboxylic acid), Tricine (≥99%; titration) were obtained from Sigma-Aldrich (Steinheim, Germany). Phosphate-buffered saline (PBS) was purchased from Gibco (Carlsbad, CA, USA). Reference substances were supplied by ChromaDex (Irvine, CA, USA), while acetonitrile, formic acid, and water were supplied for LC analysis by Merck (Darmstadt, Germany). All other chemicals were of analytical grade and were obtained from the Polish Chemical Reagent Company (POCH, Gliwice, Poland).

### 3.2. Plant Material

This study is based on the leaves and fruits of two *Cotoneaster* cultivars collected from the UMCS Botanical Garden of Maria Curie-Skłodowska University in Lublin (Poland), at altitude of 181.2 m a.s.l. (coordinates 51°15′46″ N; 22°30′51″ E) in September 2020. Taxonomical identification was confirmed by Dr. A. Cwener, an employee of the Botanical Garden who specializes in *Cotoneaster*.

*Cotoneaster nebrodensis* (Guss.) K. Koch (inventory no. 127, Figure 9) was introduced to cultivation in UMCS Botanical Garden in Lublin in 1974, whereas *C. roseus* Edgew. (inventory no. 2886, Figure 10) in 2003. The origin of described plants has been established at the University of Warsaw Botanic Garden (52°13’03.7″ N 21°01’39.2″ E) and Botanical Garden of Adam Mickiewicz University in Poznań (52°25’11.5″ N 16°52’59.4″ E), respectively.

### 3.3. Preparation of the Extracts

The leaves and fruits of the plants were washed and dried in the shade at room temperature (24 °C ± 0.5 °C) to achieve constant weight [76]. The plant material was grounded into a fine powder with the use of a grinder. To obtain extracts mixture of the methanol–acetone–water (3:1:1, *v*/*v*/*v*; 3 × 100 mL) or 60% ethanol was added and sonicated for 30 min at a controlled temperature (40 ± 2 °C). The residues were filtered, reextracted three times, and subsequently concentrated using reduced pressure, and lyophilized in a vacuum concentrator (Free Zone 1 apparatus; Labconco, Kansas City, MO, USA) in order to obtain dried residues.

### 3.4. Total Flavonoid, Phenolic, and Phenolic Acids Content

Total flavonoid (TFC) and total phenolic content (TPC) were established using colorimetric assays as described previously [25]. The absorbance was measured at 430 and 680 nm, respectively, using a Pro 200F Elisa Reader (Tecan Group Ltd., Männedorf, Switzerland). The results of TPC were expressed as mg of gallic acid equivalent (GAE) per 1 g of dry extract (DE). The results of TFC were expressed as mg of quercetin equivalent (QE) per 1 g of DE. Total phenolic acid content (TPAC) was assessed using Arnov’s reagent as described in the Polish Pharmacopoeia IX (an official translation of PhEur 7.0) [77], and the results were expressed as mg of caffeic acid equivalent (CAE) per 1 g of DE.

### 3.5. LC-MS Analysis

The chromatographic measurements were performed using a LC/MS system from Thermo Scientific (Q-EXATCTIVE and ULTIMATE 3000, San Jose, CA, USA) equipped with an ESI source. ESI was operated in positive polarity modes under the following conditions: spray voltage—4.5 kV; sheath gas—40 arb. units; auxiliary gas—10 arb. units; sweep gas—10 arb. units; and capillary temperature—320 °C. Nitrogen (>99.98%) was employed as sheath, auxiliary and sweep gas. The scan cycle used a full-scan event at a resolution of 70,000. A Gemini SYNERGI 4u Polar-RP column (250 × 4.6 mm, 5 μm) and a Phenomenex Security Guard ULTRA LC type guard column (the both from Phenomenex, Torrance, CA, USA) were employed for chromatographic separation, which was performed using gradient elution. The mobile phase A was 25 mM formic acid in water; the mobile phase B was 25 mM formic acid in acetonitrile. The gradient program started at 5% B increasing to 95% for 60 min; isocratic elution followed (95% B) next for 10 min. The total run time was 70 min at the mobile phase flow rate 0.5 mL/min. The column temperature was 25 °C. In the course of each run, the MS spectra in the range of 100–1000 *m*/*z* were collected continuously.

The amounts of the identified compounds were carried out based on the calibration curves obtained for the standard.

### 3.6. Antioxidant Activity

All antioxidant and enzyme inhibitory assays were conducted in 96-well plates (Nunclon, Nunc, Roskilde, Denmark) using Infinite Pro 200F Elisa Reader (Tecan Group Ltd., Männedorf, Switzerland). The experiments were performed in triplicate.

#### 3.6.1. DPPH^•^ Assay

The 2,2-diphenyl-1-picryl-hydrazyl (DPPH^•^) free radical scavenging activity of *Cotoneaster* extracts and the positive control—ascorbic acid (AA)—was examined using the method described previously [25], but with some modifications. After 30 min of incubation at 28 °C, the decrease in DPPH^•^ absorbance caused by the tested extracts was measured at 517 nm. The results were expressed as values of IC_50_.

#### 3.6.2. ABTS^●+^ Assay

The ABTS^●+^ decolorization assay was the second method applied for the assessment of antioxidant activity [25]. The absorbance was measured at 734 nm. Trolox was used as a positive control. The results were expressed as values of IC_50_.

#### 3.6.3. Metal Chelating Activity (CHEL)

The metal chelating activity was established using the method described by Guo et al. [78], modified in our previous study [25]. The absorbance was measured at 562 nm. As a positive control, Na_2_EDTA*2H_2_O was used. The results were expressed as the IC_50_ values of the *Cotoneaster* extracts based on concentration–inhibition curves.

### 3.7. Enzyme Inhibitory Activity

#### 3.7.1. Cyclooxygenase-1 (COX-1) and Cyclooxygenase-2 (COX-2) Inhibitory Activity

The extracts of the *Cotoneaster* species were examined for cyclooxygenase-1 (COX-1) and cyclooxygenase-2 (COX-2) inhibitory activity using a COX (ovine/human) Inhibitor Screening Assay Kit (Cayman Chemical, MI, USA) according to the protocol of the manufacturer. The extracts were tested at different concentrations. Indomethacin was used as a positive control.

#### 3.7.2. Lipoxygenase Inhibitory Activity

The anti-lipoxygenase activity of the *Cotoneaster* extracts was determined using the Lipoxygenase Inhibitor Screening Assay Kit (Cayman Chemical, MI, USA) according to the protocol of the manufacturer. The extracts were tested at different concentrations. The effective concentration (μg/mL) in which lipoxygenase activity is inhibited by 50% (IC_50_) was estimated graphically. Indomethacin was used as a positive control.

#### 3.7.3. Hyaluronidase Inhibitory Activity

The anti-hyaluronidase activity was established using the method described by Liyanaarachchi et al. [79]. After 20 min incubation at 37 °C, the absorbance was measured at 585 nm. The extracts were tested at different concentrations. Indomethacin was used as a positive control.

### 3.8. Antibacterial Activity

The antimicrobial activity of *Cotoneaster* extracts was tested in vitro against the following strains: microaerobic Gram-positive *Propionibacterium granulosum* PCM 2462, *Cutibacterium acnes* PCM 2334, *C. acnes* PCM 2400 (bought as *Propionibacterium* at the Hirszfeld Institute of Immunology and Experimental Therapy, PAN, Poland) and aerobic Gram-positive *Staphylococcus aureus* ATCC 25923, *S. epidermidis* ATCC 12228 and aerobic Gram-negative *Escherichia coli* ATCC 25992 bacterial strains. For the antibacterial activity determination, Mueller–Hinton agar or broth (MH-agar, MH-broth) for aerobic and Brain-Heart Infusion agar or broth (BHI-agar, BHI-broth) for microaerobic strains were used. The inoculum was prepared by subculturing bacteria in MH-agar or BHI-agar at 37 °C for 24 h or 48 h, respectively. Next, the inocula were prepared with fresh microbial cultures in sterile 0.9% NaCl to 0.5 McFarland turbidity standard, 1.5 × 10^8^ CFU/mL (CFU: colony-forming unit).

#### 3.8.1. Disc Diffusion Assay in Solid Medium

The antibacterial activity of all extracts was performed by a modified agar disc diffusion method based on the Kirby–Bauer procedure [80]. The bacterial inoculum was spread (using a cotton swab) on the surface of the Petri dishes containing appropriate agar. The stock solutions of all tested compounds (10 mg/mL) were prepared using DMSO. Next, 100 µg of the extracts were placed on inoculated Petri dishes. Plates with MH-agar (for aerobic strains) were incubated at 37 °C for 24 h and plates with BHI-agar (for microaerobic strains) at 37 °C for 48 h. The diameter of the growth inhibition zone around each sample was measured after incubation using a microbiological ruler.

#### 3.8.2. MIC Determination

The minimum inhibitory concentration (MIC) of the *Cotoneaster* extracts was determined for the bacterial strains that exhibited the bacterial growth inhibition zones. The test was performed using the double serial microdilution in the 96-well microtiter plates according to CLSI method with some modifications (CLSI Performance standards for antimicrobial susceptibility testing, 2008. Eighteenth International Supplement. CLSI document M7-MIC. Clinical Laboratory Standards Institute, Wayne). The appropriate broth (200 µL) was pipetted into each well. Double serial dilution of tested derivatives was performed in the test wells, causing concentrations ranging from 4000 µg/mL to 250 µg/mL. Finally, 2 µL of tested bacteria inoculum were added to the wells (except for negative sterility control). The tests were performed either at 37 °C for 24 h (aerobic strains) or 48 h (microaerobic strains). After incubation, the panel was digitally analyzed at 600 nm using the microplate reader Bio Tech Synergy (USA) with a proprietary software system. The growth intensity in each well was compared with the negative and positive controls.

### 3.9. Cytotoxic Activity

The cytotoxicity assessment of the extracts was performed on a BJ cell line (normal human fibroblasts, ATCC CRL-2522^TM^) in accordance with the protocol described by our research team previously [75]. The BJ cells were seeded in 96-well plates, and then incubated for 24 h at 37 °C (5% CO_2_, 95% air). On the next day, the culture medium was replaced with two-fold serial dilutions of tested extracts. After 24-h incubation, the cell viability was evaluated using a MTT assay. The obtained data were presented as mean values ± standard deviations (SD). These results were subjected to four-parameter nonlinear regression analyses (GraphPad Prism 5, version 5.04, GraphPad Software, San Diego, CA, USA) in order to determine values of half-maximum cytotoxic concentration (CC_50_). Moreover, the therapeutic indexes (TIs) were determined. The TIs denoted the potential safety of the extracts in vitro and were calculated as a ratio of CC_50_ and MIC [75]. The higher the TIs value, the greater safety of the extracts for mammalian cells.

### 3.10. Statistical Analysis

Statistical analyses were performed by using Origin Statistica 13.3 (StatSoft, Cracow, Poland). All trials were performed in triplicate to ensure their exactness. One-way ANOVA followed by Tukey’s multiple comparison test (*p* < 0.05) was employed to estimate statistically significant differences among the means. Before carrying out ANOVA, the assumptions for normality of data distribution with the use of Shapiro–Wilk’s tests were verified. Additionally, the normality of data was tested using Kolmogorov–Smirnov test. Levene’s test was employed to check the homogeneity of variances. Means (±SD) with *p* < 0.05 were considered statistically different. A hierarchical cluster analysis (HCA) was performed for all data obtained as a result of performed experiments. The Ward’s method and Euclidean squared distance were employed.

## 4. Conclusions

In recent years, a fast-moving shift in recognition of acne was observed. Currently, acne is rather considered an inflammatory condition, unlike a contagious disorder [81]. In this context, there is a need to reduce the use of antibiotics and to look for non-antibiotic alternatives.

Herbal medicines that contain active compounds with proven antioxidant, anti-inflammatory, and antimicrobial properties are of particular importance as they can replace or support conventional therapies possessing remarkable side effects [8]. Plants play a pivotal role as a source of biologically active natural compounds and may contribute to restricting the use of synthetic substances, especially antibiotics. According to available research [82], polyphenolic extracts seem to be the most effective with respect to this purpose.

In our in vitro study, we characterized the composition and evaluated the biological activities of the extracts from *C. nebrodensis* and *C. roseus*. As a consequence, we identified the main compounds presenting in extracts as well as determined the antioxidant, anti-inflammatory, antimicrobial, and cytotoxic properties of such extracts. We proved that both species are rich in chlorogenic acid, isoquercitrin, (+)-catechin, and cinnamic acid, which are most likely responsible for their beneficial biological activities. The applicability of *Cotoneaster* species in the treatment of skin disorders reported by traditional medicine [24] is provided by numerous polyphenolic components, and our scrutiny partially explains this fact. According to Soleymani et al. [8], the vast majority of secondary plant metabolites examined in the acne treatment were from the category of polyphenols; therefore, *C. roseus* as well as *C. nebrodensis* appear to be promising candidates for future research, as the content of the phenolic compounds is significant.

The present investigation provides new insight into the possibility of applying the extracts obtained from untested species, namely *C. roseus* and *C. nebrodensis,* in the treatment of inflammation-related diseases, including acne, in the adjunctive or prophylaxis therapy. The obtained results indicated that especially methanol–acetone–water (3:1:1, *v*/*v*) extracts from the leaves of *C. roseus* and fruits of *C. nebrodensis* showed relatively high antioxidant, anti-inflammatory, and antibacterial properties in comparison with the positive standards. It is especially worth noting that the extract obtained from the fruits of *C. nebrodensis* (CNMF) seems to be the most promising further anti-acne agent because, apart from antioxidant and anti-inflammatory activities, it possessed the best therapeutic indexes, which potentially determines its therapeutic safety in the term of antibacterial and cytotoxic activities.

Thus, further in vivo clinical and animal studies of the tested extracts are required to evaluate their modes of action and potential side effects. Antioxidants, as well as pro-inflammatory activity, should be confirmed by chemical and human plasma models [16]. The extract obtained from the fruits of *C. nebrodensis* seems to be an especially promising agent for the treatment of acne vulgaris, and for this reason, it will be subjected to further in vivo study to confirm its “multidirectional” activity.

## Figures and Tables

**Figure 1 molecules-27-02907-f001:**
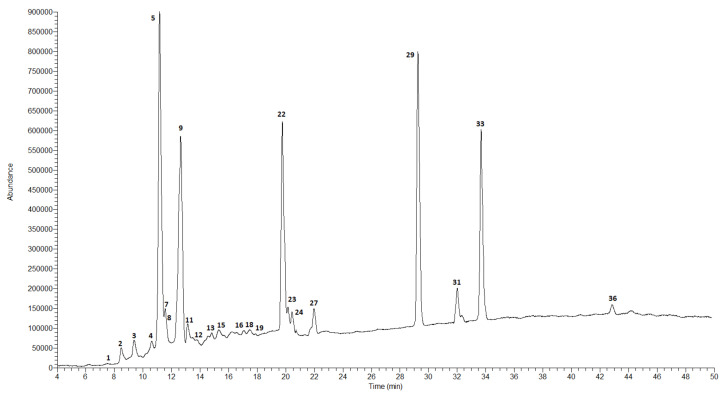
LC-MS chromatogram in SCAN mode for the methanol–acetone–water extract from the fruits of *C. nebrodensis*.

**Figure 2 molecules-27-02907-f002:**
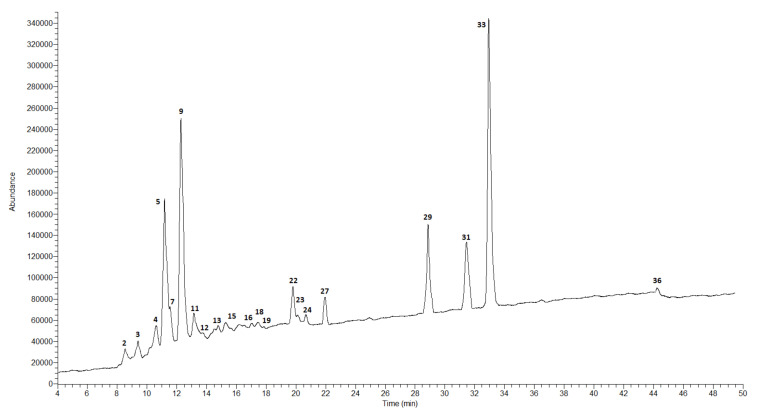
LC-MS chromatogram in SCAN mode for 60% ethanol extract from the fruits of *C. nebrodensis*.

**Figure 3 molecules-27-02907-f003:**
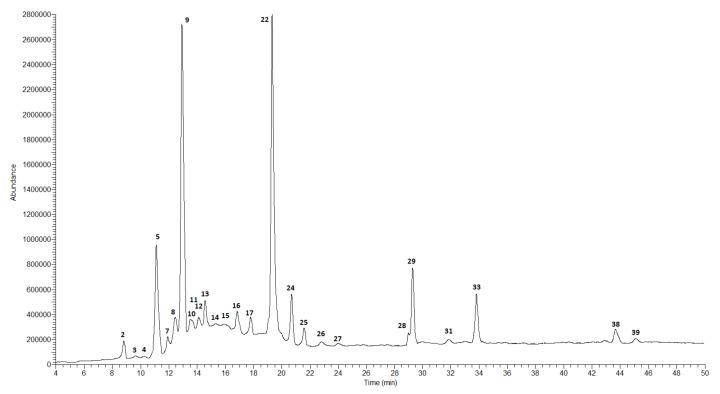
LC-MS chromatogram in SCAN mode for the methanol–acetone–water extract from the fruits of *C. roseus*.

**Figure 4 molecules-27-02907-f004:**
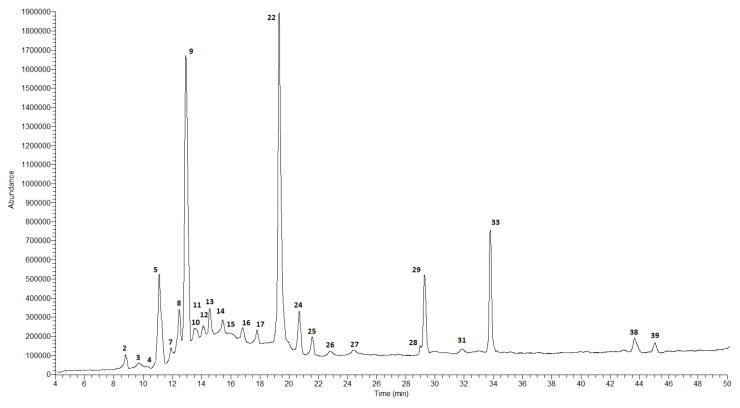
LC-MS chromatogram in SCAN mode for 60% ethanol extract from the fruits of *C. roseus*.

**Figure 5 molecules-27-02907-f005:**
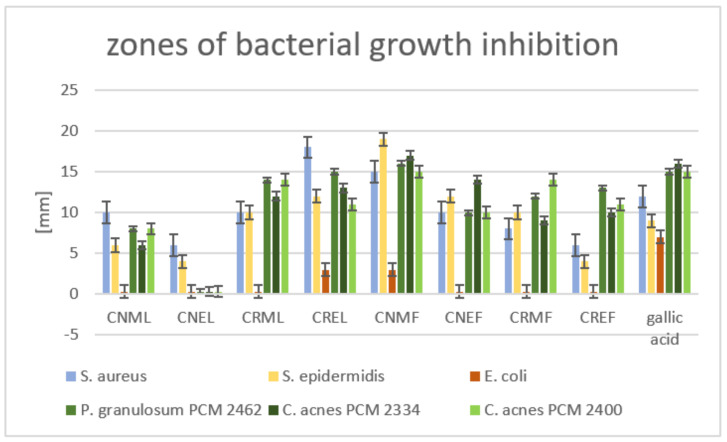
Zones of bacterial growth inhibition caused by extracts obtained from the fruits or leaves of *C. roseus* and *C. nebrodensis*. CNML—methanol–acetone–water (3:1:1, *v*/*v*) extract of the leaves of *C. nebrodensis*; CNEL—60% ethanol extract of the leaves of *C. nebrodensis*; CRML—methanol–acetone–water (3:1:1, *v*/*v*) extract of the leaves of *C. roseus*; CREL—60% ethanol extract of the leaves of *C. roseus*; CNMF—methanol–acetone–water (3:1:1, *v*/*v*) extract of the fruits of *C. nebrodensis*; CNEF—60% ethanol extract of the fruits of *C. nebrodensis*; CRMF—methanol–acetone–water (3:1:1, *v*/*v*) extract of the fruits of *C. roseus*; CREF—60% ethanol extract of the fruits of *C. roseus*.

**Figure 6 molecules-27-02907-f006:**
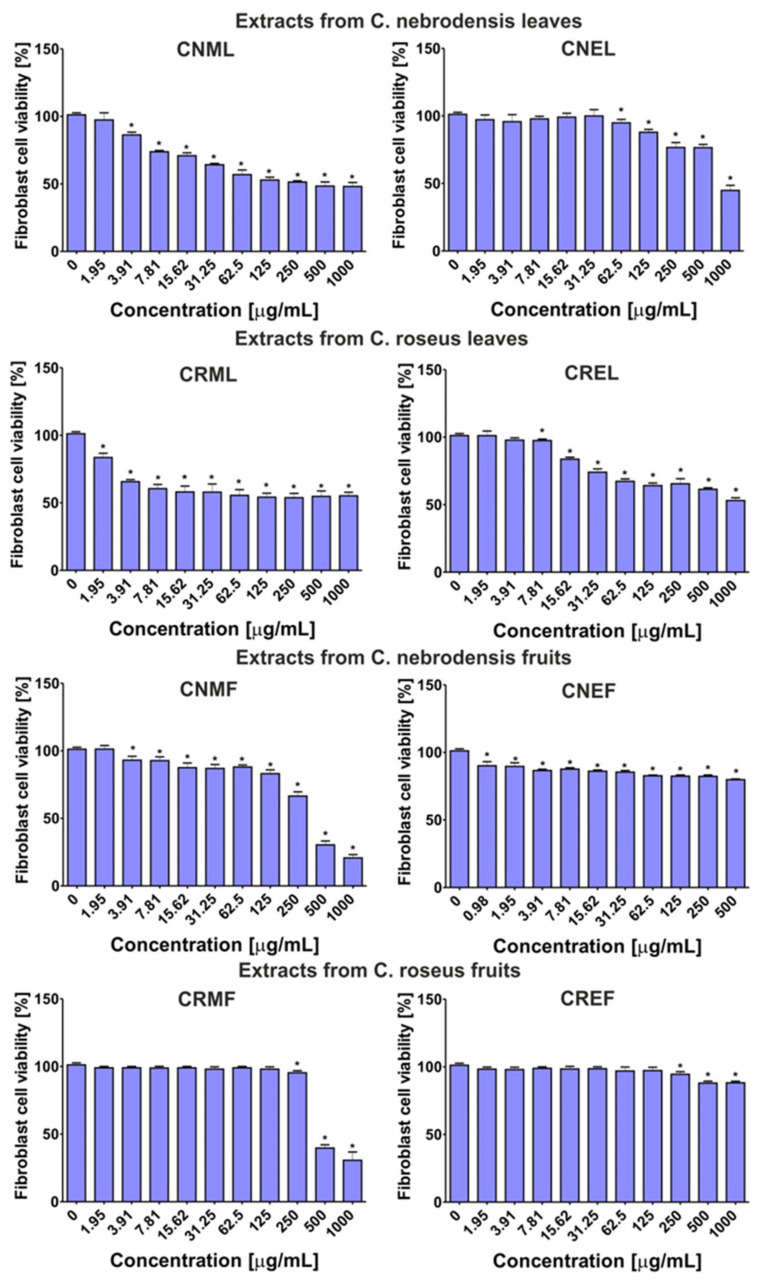
Viability of human normal fibroblasts (BJ cell line, ATCC CRL-2522^TM^) after 24-h treatment with serial dilutions of the extracts obtained from the leaves and fruits of *C. nebrodensis* (CNML, CNEL and CNMF, CNEF) and *C. roseus* (CRML, CREL and CRMF, CREF). The cell viability was evaluated using a MTT assay. * Significantly different data (*p* < 0.05, unpaired *t*-test) compared to control, i.e., culture medium without substances—0 μg/mL. The CNEF extract was evaluated at the concentrations of 500–0.98 μg/mL due to its low solubility.

**Figure 7 molecules-27-02907-f007:**
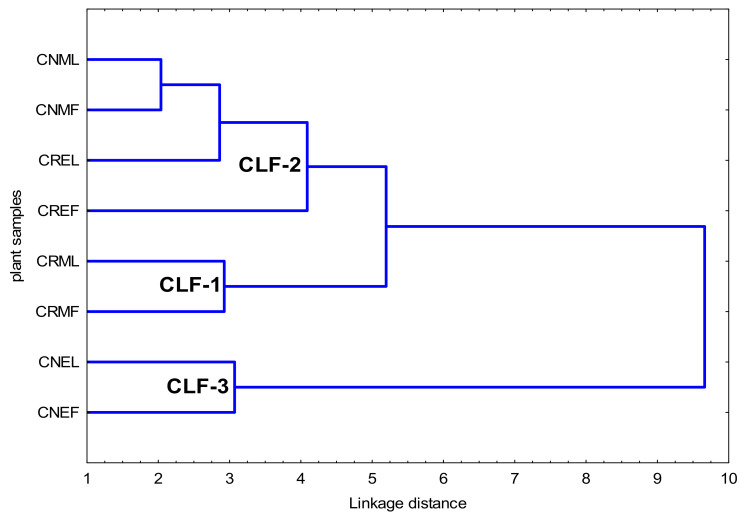
Hierarchical cluster analysis with all variables obtained for the *Cotoneaster* leaves and fruits using the Ward’s method and Euclidean squared distance. CNML—methanol–acetone–water (3:1:1, *v*/*v*) extract of the leaves of *C. nebrodensis*; CNEL—60% ethanol extract of the leaves of *C. nebrodensis*; CRML—methanol–acetone–water (3:1:1, *v*/*v*) extract of the leaves of *C. roseus*; CREL—60% ethanol extract of the leaves of *C. roseus*; CNMF—methanol–acetone–water (3:1:1, *v*/*v*) extract of the fruits of *C. nebrodensis*; CNEF—60% ethanol extract of the fruits of *C. nebrodensis*; CRMF—methanol–acetone–water (3:1:1, *v*/*v*) extract of the fruits of *C. roseus*; CREF—60% ethanol extract of the fruits of *C. roseus*; CLF-1—cluster leaves fruits 1, CLF-2—cluster leaves fruits 2, CLF-3—cluster leaves fruits 3.

**Figure 8 molecules-27-02907-f008:**
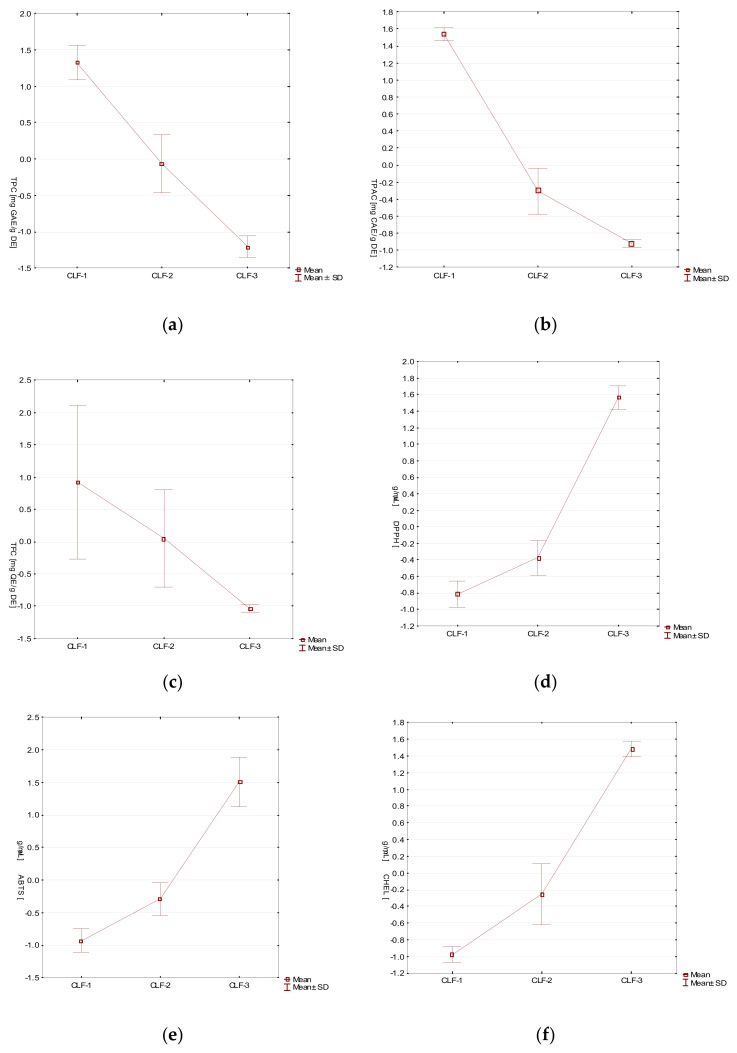
Box-whisker plot analysis of: (**a**) TPC, (**b**) TPAC, (**c**) TFC, (**d**) DPPH, (**e**) ABTS, (**f**) CHEL, (**g**) LPO, (**h**) HYAL, (**i**) COX-1, (**j**) COX-2 in the Cotoneaster leaves and fruits samples. Mean values ± standard deviation (SD).

**Figure 9 molecules-27-02907-f009:**
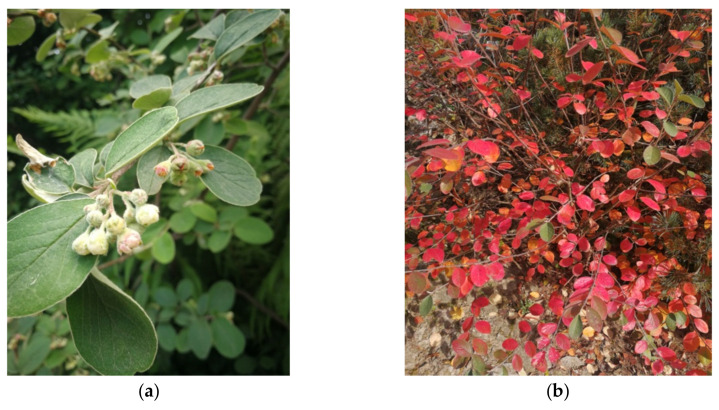
*C. nebrodensis* in the UMCS Botanical Garden of Maria Curie-Skłodowska University in Lublin taken in May (**a**) and in October (**b**).

**Figure 10 molecules-27-02907-f010:**
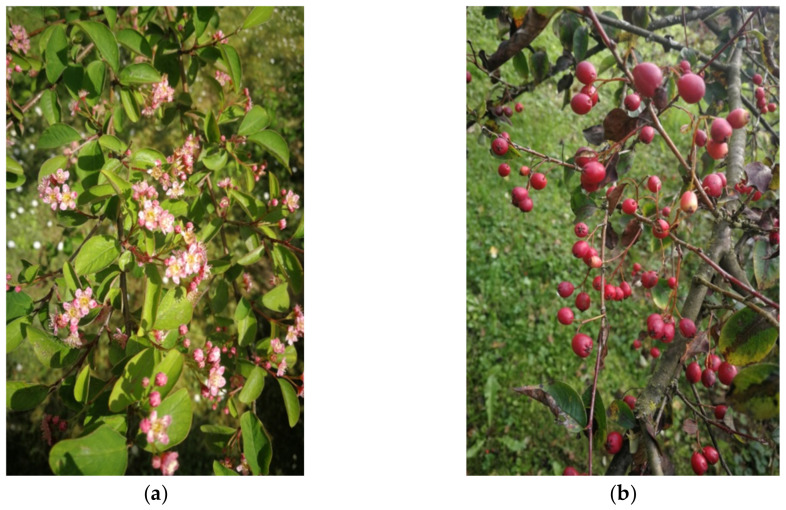
*C. roseus* in the UMCS Botanical Garden of Maria Curie-Skłodowska University in Lublin taken in May (**a**) and in September (**b**).

**Table 1 molecules-27-02907-t001:** The total content of phenolic (TPC), flavonoid (TFC), and phenolic acids (TPAC) in the leaves and fruits of *C. nebrodensis* and *C. roesus*.

Sample	Extraction Yield (% DE)	Total Phenolic Content [mg GAE/g DE]	Total Phenolic Acids [mg CAE/g DE]	Total Flavonoid Content [mg QE/g DE]
CNML	54.0	49.55 ± 0.11 ^a^	10.39 ± 0.34 ^a^	17.21 ± 0.29 ^a^
CNEL	4.75	19.50 ± 0.20 ^b^	2.28 ± 0.12 ^b^	3.53 ± 0.09 ^b^
CRML	44.7	118.43 ± 0.41 ^c^	57.41 ± 0.39 ^c^	51.60 ± 0.71 ^c^
CREL	1.4	88.59 ± 0.70 ^d^	14.07 ± 0.09 ^d^	37.58 ± 0.70 ^d^
CNMF	19.5	53.60 ± 0.16 ^e^	18.43 ± 0.60 ^e^	25.40 ± 0.32 ^e^
CNEF	1.5	10.45 ± 0.55 ^f^	3.67 ± 0.17 ^f^	1.90 ± 0.32 ^f^
CRMF	19.7	132.45 ± 0.21 ^g^	59.79 ± 0.42 ^g^	22.03 ± 0.14 ^g^
CREF	2.9	68.20 ± 0.26 ^h^	24.76 ± 0.10 ^h^	6.50 ± 0.40 ^h^

DE—dry extract; GAE—Gallic Acid Equivalent; CAE—Caffeic Acid Equivalent; QE—Quercetin Equivalent; CNML—methanol–acetone–water (3:1:1, *v*/*v*) extract of the leaves of *C. nebrodensis*; CNEL—60% ethanol extract of the leaves of *C. nebrodensis*; CRML—methanol–acetone–water (3:1:1, *v*/*v*) extract of the leaves of *C. roseus*; CREL—60% ethanol extract of the leaves of *C. roseus*; CNMF—methanol–acetone–water (3:1:1, *v*/*v*) extract of the fruits of *C. nebrodensis*; CNEF—60% ethanol extract of the fruits of *C. nebrodensis*; CRMF—methanol–acetone–water (3:1:1, *v*/*v*) extract of the fruits of *C. roseus*; CREF—60% ethanol extract of the fruits of *C. roseus*. Values were presented as mean ± standard deviation (n = 3). The letters indicate whether a significant difference among the samples existed. The same letters mean no significant difference and different letters mean significant difference. One-Way ANOVA test, followed by Tukey’s multiple comparison tests, *p* < 0.05.

**Table 2 molecules-27-02907-t002:** Content of active compounds (tentatively determined) in the leaves of *C. nebrodensis* (CNML—methanol–acetone–water (3:1:1) extract, CNEL—60% ethanol extract) and *C. roseus* (CRML—methanol–acetone–water (3:1:1) extract, CREL—60% ethanol extract). LOQ—limit of quantification; DE—dry extract.

No	Compound	Calibration Standard	Amounts [μg/g DE]			
			CNML		CRML	
			22,388.6	CNEL	87,006.2	CREL
			2.5x	9057.4	1.8x	48,141.6
1	mannitol	glucose	<LOQ	<LOQ	1189.3 ± 41.0	<LOQ
2	quercetin 3-*O*-rutinoside (rutin)	rutin	559.2 ± 34.6	85.6 ± 1.1	3443.0 ± 143.3	1095.1 ± 64.2
3	quercetin 3-*O*-(2″-*O*-xylosyl)galactoside	rutin	436.2 ± 19.0	379.5 ± 9.8	580.2 ± 23.1	421.9 ± 18.6
4	quercetin 3-*O*-gentiobioside	rutin	90.0 ± 5.1	<LOQ	369.3 ± 19.1	92.8 ± 2.4
5	vitexin 2″-*O*-arabinoside	rutin	118.2 ± 4.9	<LOQ	2867.4 ± 117.0	1869.5 ± 75.3
6	apigenin 6,8-*C*-dicelobioside	rutin	633.5 ± 19.2	173.0 ± 5.3	1316.2 ± 49.0	518.4 ± 16.6
7	vitexin 2″-*O*-rhamnoside	rutin	343.9 ± 16.3	84.2 ± 2.7	675.8 ± 29.1	196.0 ± 4.1
8	quercetin 3-*O*-glucoside (isoquercitrin)	rutin	1064.5 ± 43.1	852.5 ± 42.5	6520.1 ± 234.0	4991.3 ± 174.8
9	quercetin 3-*O*-galactoside (hyperoside)	rutin	709.1 ± 29.8	487.9 ± 20.3	4141.0 ± 152.1	3086.5 ± 126.0
10	kaempferol 3-*O*-glucoside (astragalin)	rutin	1468.7 ± 49.1	1170.0 ± 42.1	991.0 ± 45.4	894.7 ± 38.5
11	quercetin 3-*O*-rhamnoside (quercitrin)	rutin	1734.0 ± 76.2	1274.5 ± 21.5	2073.9 ± 77.1	1810.2 ± 46.9
12	7-methylkaempferol 4′-*O*-glucoside	rutin	529.4 ± 29.0	346.2 ± 17.5	1337.5 ± 42.0	963.2 ± 49.7
13	3′,4′-dihydroxy-6-methoxyflavone 7-*O*-rhamnoside	rutin	680.9 ± 27.3	294.6 ± 13.7	1925.3 ± 63.2	1049.0 ± 54.8
14	apigenin 8-*C*-glucoside (vitexin)	rutin	85.2 ± 4.0	<LOQ	941.4 ± 39.0	261.7 ± 5.4
15	apigenin 7-*O*-glucoside	rutin	959.1 ± 34.2	490.8 ± 23.6	2457.0 ± 84.5	1864.2 ± 43.7
16	biochanin A 7-*O*-glucoside (sissotrin)	rutin	307.0 ± 9.9	83.3 ± 1.8	935.1 ± 34.0	612.4 ± 24.0
17	5,7,2′,5′-tetrahydroxyflavanone 7-*O*-glucoside	rutin	690.2 ± 25.1	472.9 ± 19.8	1849.7 ± 61.2	129.0 ± 14.7
18	5-methylgenistein 4′-*O*-glucoside	rutin	469.6 ± 17.0	117.1 ± 2.9	1580.5 ± 52.1	974.2. ± 5.8
19	orbicularin	quercetin	456.9 ± 13.2	94.5 ± 4.6	1537.9 ± 43.5	799.3 ± 32.4
20	*p*-hydroxybenzoic acid	*p*-hydroxybenzoic acid	119.1 ± 5.1	92.0 ± 2.1	<LOQ	<LOQ
21	benzoic acid	benzoic acid	410.8 ± 12.9	216.4 ± 5.7	<LOQ	<LOQ
22	gentisic acid	gentisic acid	<LOQ	< LOQ	<LOQ	<LOQ
23	protocatechuic acid	protocatechuic acid	<LOQ	< LOQ	<LOQ	<LOQ
24	syringic acid	syringic acid	308.7 ± 13.1	274.0 ± 5.2	834.7 ± 38.2	649.2 ± 8.7
25	vanillic acid	vanillic acid	189.0 ± 7.0	<LOQ	113.2 ± 4.0	<LOQ
26	caffeoylmalic acid	caffeic acid	402.2 ± 14.5	<LOQ	719.4 ± 29.0	<LOQ
27	chlorogenic acid	chlorogenic acid	1455.7 ± 51.8	1265.0 ± 42.8	26,836.5 ± 987.0	21,822.0 ± 584.0
28	prunasin	glucose	93.0 ± 4.1	24.4 ± 0.9	1145.7 ± 41.2	238.0 ± 10.7
29	*p*-coumaric acid	*p*-coumaric acid	<LOQ	<LOQ	264.0 ± 9.3	129.5 ± 1.8
30	amygdalin	glucose	<LOQ	<LOQ	538.6 ± 23.0	<LOQ
31	caffeic acid	caffeic acid	67.2 ± 3.0	28.1 ± 2.5	543.9 ± 19.2	318.6 ± 17.1
32	cinnamic acid	cinnamic acid	98.0 ± 4.1	12.4 ± 0.3	175.6 ± 6.0	74.0 ± 0.9
33	ferulic acid	ferulic acid	431.9 ± 16.2	193.0 ± 6.6	1907.8 ± 61.1	586.4 ± 12.1
34	salicylic acid	salicylic acid	107.5 ± 4.0	<LOQ	<LOQ	<LOQ
35	7,8-dimethoxy-6-hydroxycoumarin	umbelliferone	97.3 ± 4.1	<LOQ	453.5 ± 18.3	<LOQ
36	cotonoate A	benzoic acid	167.8 ± 7.4	29.3 ± 0.8	1034.0 ± 42.1	286.7 ± 19.5
37	horizontoate A	benzoic acid	80.0 ± 3.9	<LOQ	1175.2 ± 49.3	795.8 ± 28.4
38	3,3′,4′-tri-*O*-methylellagic acid	quercetin	123.1 ± 5.0	18.5 ± 1.2	1937.1 ± 72.0	853.5 ± 32.1
39	scopoletin	umbelliferone	4143.5 ± 152.2	<LOQ	8880.0 ± 331.2	<LOQ
40	arbutin	glucose	883.0 ± 30.8	<LOQ	237.5 ± 12.0	<LOQ
41	5-methylgenistein	quercetin	184.3 ± 5.0	<LOQ	464.9 ± 16.1	<LOQ
42	quercetin	quercetin	137.7 ± 4.7	106.0 ± 2.6	290.0 ± 8.5	231.2 ± 4.3
43	horizontoate C	oleic acid	679.5 ± 25.1	57.9 ± 3.2	1743.0 ± 63.6	265.0 ± 4.7
44	eriodictyol	quercetin	268.0 ± 13.0	186.5 ± 4.8	359.9 ± 14.1	174.4 ± 6.5
45	5,7,2′,5′-tetrahydroxyflavanone	quercetin	210.2 ± 8.5	<LOQ	382.1 ± 13.0	<LOQ
46	naringenin	quercetin	395.7 ± 15.0	147.3 ± 3.8	238.0 ± 11.5	87.8 ± 1.8

**Table 3 molecules-27-02907-t003:** Content of the active compounds (tentatively determined) in the fruits of *C. nebrodensis* (CNMF—methanol–acetone–water (3:1:1) extract, CNEF—60% ethanol extract) and *C. roseus* (CRMF—methanol–acetone–water (3:1:1) extract, CREF—60% ethanol extract). LOQ—limit of quantification; DE—dry extract.

No	Compounds	Calibration Standard	Amounts [μg/g DE]			
			CNMF	CNEF	CRMF	CREF
1	Gallic acid	Gallic acid	13.4 ± 0.6	7.5 ± 0.4	107.9 ± 5.1	19.4 ± 0.8
2	3-*O*-Caffeoylquinic acid	3-*O*-Caffeoylquinic acid	233.2 ± 9.5	210.8 ± 8.6	2801.1 ± 112.6	1189.0 ± 58.1
3	Vanillic acid hexoside	Vanillic acid	235.5 ± 10.6	137.6 ± 5.7	411.6 ± 16.8	285.4 ± 13.9
4	Syringic acid hexoside	Syringic acid	165.4 ± 6.7	40.8 ± 1.9	271.3 ± 11.4	222.6 ± 8.9
5	(+)-Catechin	(+)-Catechin	8940.7 ± 377.2	1006.1 ± 42.1	6936.0 ± 302.4	1063.3 ± 44.8
6	Procyanidin C-1	(-)-Epicatechin	3.4 ± 0.2	2.8 ± 0.1	314.8 ± 13.9	14.3 ± 0.6
7	Protocatechuic acid	Protocatechuic acid	491.6 ± 20.1	352.9 ± 14.5	827.0 ± 35.1	689.8 ± 31.6
8	Procyanidin B-2	(-)-Epicatechin	1372.0 ± 58.2	18.5 ± 0.8	1974.1 ± 90.2	93.0 ± 4.3
9	5-*O*-caffeoylquinic acid (chlorogenic acid)	5-*O*-caffeoylquinic acid (chlorogenic acid)	5907.0 ± 294.2	1703.5 ± 80.9	24,124.0 ± 1153.1	14,519.8 ± 661.5
10	*p*-Hydroxybenzoic acid	*p*-Hydroxybenzoic acid	567.8 ± 25.2	407.8 ± 19.0	663.9 ± 27.7	583.1 ± 26.0
11	(-)-Epicatechin	(-)-Epicatechin	438.3 ± 20.0	384.9 ± 19.0	1109.0 ± 50.7	205.4 ± 9.6
12	4-*O*-Caffeoylquinic acid	4-*O*-Caffeoylquinic acid	968.0 ± 46.4	59.8 ± 2.7	3849.1 ± 160.1	1593.0 ± 76.3
13	Caffeic acid hexoside	Caffeic acid	289.3 ± 13.7	33.4 ± 1.5	1139.5 ± 51.0	560.2 ± 22.7
14	Caffeic acid	Caffeic acid	3.3 ± 0.1	2.4 ± 0.1	716.5 ± 30.5	453.5 ± 18.8
15	Syringic acid	Syringic acid	54.5 ± 2.7	38.5 ± 1.7	23.3 ± 1.1	18.0 ± 0.7
16	*p*-Coumaric acid	*p*-Coumaric acid	72.4 ± 3.5	49.5 ± 2.3	750.8 ± 33.1	349.5 ± 15.8
17	*o*-Coumaric acid	*p*-Coumaric acid	64.8 ± 2.8	41.9 ± 1.8	838.4 ± 37.8	529.7 ± 22.6
18	Vanilin	Vanilin	242.4 ± 10.3	122.3 ± 5.2	34.7 ± 1.5	28.5 ± 1.4
19	Ferulic acid	Ferulic acid	128.1 ± 5.2	107.9 ± 4.9	152.8 ± 7.1	151.3 ± 6.4
20	3,5-Di-*O*-caffeoylquinic acid	3,5-Di-*O*-caffeoylquinic acid	62.1 ± 2.9	53.7 ± 2.3	88.8 ± 3.8	51.1 ± 2.3
21	Querectin-3-*O*-β-D-(6″-*O*-α-L-rhamnosyl)glucoside (rutin)	Rutin	27.7 ± 1.1	7.6 ± 0.3	160.4 ± 7.0	93.4 ± 4.4
22	3-*O-p*-Coumaroylquinic acid	3-*O*-Caffeoylquinic acid	4268.3 ± 205.3	384.9 ± 17.4	27,744.0 ± 1315.1	18,178.4 ± 892.6
23	Quercetin 3-*O*-β-D-(2″-*O*-β-D-xylosyl)galactoside	Rutin	149.8 ± 6.8	38.5 ± 1.8	118.9 ± 5.9	71.7 ± 3.3
24	Quercetin 3-*O*-β-D-galactoside (hyperoside)	Rutin	262.2 ± 10.6	91.8 ± 4.3	2393.3 ± 96.9	1406.3 ± 56.4
25	5-*O-p*-Coumaroylquinic acid	5-*O*-Caffeoylquinic acid	133.0 ± 6.1	46.5 ± 2.0	3125.0 ± 135.9	2618.1 ± 114.4
26	Quercetin 3-*O*-β-D glucoside (isoquercitrin)	Rutin	108.2 ± 4.8	57.6 ± 2.3	377.7 ± 15.4	295.4 ± 13.3
27	Quercetin 3-*O*-α-L rhamnoside (quercitrin)	Rutin	769.8 ± 36.1	317.1 ± 15.2	644.1 ± 28.0	579.3 ± 28.0
28	Hesperidin	Rutin	25.3 ± 1.1	4.2 ± 0.2	670.7 ± 30.2	376.5 ± 16.2
29	Naringin	Rutin	8917.7 ± 409.3	455.6 ± 19.2	4497.0 ± 190.2	1673.0 ± 75.5
30	Biochanin-7-*O*-glucoside	Rutin	7.4 ± 0.3	4.5 ± 0.2	66.3 ± 2.8	53.7 ± 2.3
31	Rosmarinic acid	Caffeic acid	349.5 ± 14.9	252.7 ± 10.8	317.1 ± 14.3	190.2 ± 8.1
32	Sinapic acid	Ferulic acid	63.3 ± 2.7	56.0 ± 2.5	22.8 ± 1.0	16.4 ± 0.8
33	Cinnamic acid	Cinnamic acid	5411.6 ± 246.8	3532.8 ± 151.2	2846.8 ± 120.4	2324.7 ± 97.9
34	Quercetin	Quercetin	38.1 ± 1.6	14.7 ± 0.6	150.5 ± 6.6	120.8 ± 5.5
35	Kaempferol	Kaempferol	13.5 ± 0.6	7.1 ± 0.3	139.5 ± 6.9	65.9 ± 3.1
36	Eriodictyol	Luteolin	304.1 ± 13.0	157.8 ± 7.2	97.2 ± 7.0	65.6 ± 2.9
37	Luteolin	Luteolin	11.6 ± 0.5	5.8 ± 0.3	82.7 ± 3.5	68.2 ± 3.1
38	Apigenin	Apigenin	59.1 ± 2.9	3.5 ± 0.2	979.4 ± 43.2	392.5 ± 18.6
39	Biochanin	Apigenin	4.8 ± 0.2	2.9 ± 0.1	678.4 ± 28.6	575.5 ± 25.1

**Table 4 molecules-27-02907-t004:** The IC_50_ values determined in antioxidant tests.

Sample	IC_50_		
	DPPH [μg/mL]	ABTS [μg/mL]	CHEL [μg/mL]
CNML	55.60 ± 0.04 ^a^	53.67 ± 0.73 ^a^	108.89 ± 0.02 ^a^
CNEL	117.79 ± 0.02 ^b^	100.12 ± 0.02 ^b^	196.01 ± 0.13 ^b^
CRML	32.12 ± 0.19 ^c^	21.04 ± 0.11 ^c^	38.33 ± 0.22 ^c^
CREL	44.12 ± 0.04 ^d^	31.98 ± 0.17 ^d^	57.99 ± 0.16 ^d^
CNMF	43.37 ± 0.09 ^e^	43.03 ± 0.13 ^e^	98.65 ± 0.29 ^e^
CNEF	125.63 ± 0.02 ^f^	121.06 ± 0.21 ^f^	205.04 ± 0.30 ^f^
CRMF	22.94 ± 0.20 ^g^	10.89 ± 0.11 ^g^	29.62 ± 0.23 ^g^
CREF	35.49 ± 0.50 ^h^	33.71 ± 0.15 ^h^	65.44 ± 0.42 ^h^
quercetin	2.38 ± 0.11	3.61 ± 0.10	6.85 ± 0.23
AA	4.29 ± 0.09	1.68 ± 0.05	nt
Trolox	3.74 ± 0.15	1.45 ± 0.02	nt
Na_2_EDTA*2H_2_O	nt	nt	4.69 ± 0.17

Data were expressed as mean values ± SD, *n* = 3. AA—ascorbic acid; Na_2_EDTA*2H_2_O—ethylenediaminetetraacetic acid, disodium dihydrate; nt—not tested; CNML—methanol–acetone–water (3:1:1, *v*/*v*) extract of the leaves of *C. nebrodensis*; CNEL—60% ethanol extract of the leaves of *C. nebrodensis*; CRML—methanol–acetone–water (3:1:1, *v*/*v*) extract of the leaves of *C. roseus*; CREL—60% ethanol extract of the leaves of *C. roseus*; CNMF—methanol–acetone–water (3:1:1, *v*/*v*) extract of the fruits of *C. nebrodensis*; CNEF—60% ethanol extract of the fruits of *C. nebrodensis*; CRMF—methanol–acetone–water (3:1:1, *v*/*v*) extract of the fruits of *C. roseus*; CREF—60% ethanol extract of the fruits of *C. roseus*. Values were presented as mean ± standard deviation (n = 3). The letters indicate whether a significant difference among the samples existed. The same letters mean no significant difference and different letters mean significant difference. The One-Way ANOVA test, followed by Tukey’s multiple comparison test, *p* < 0.05.

**Table 5 molecules-27-02907-t005:** Anti-lipoxygenase, anti-hyaluronidase, and anti-cyclooxygenase activities of the leaves and fruits of *C. nebrodensis* and *C. roseus*.

Sample	IC_50_ [µg/mL]			
	Lipoxygenase Inhibition	Hyaluronidase Inhibition	COX-1 Inhibition	COX-2 Inhibition
CNML	213.98 ± 0.17 ^a^	38.55 ± 0.45 ^a^	63.26 ± 0.19 ^a^	51.01 ± 0.24 ^a^
CNEL	324.95 ± 0.15 ^b^	58.20 ± 0.04 ^b^	14.31 ± 0.38 ^b^	23.68 ± 0.27 ^b^
CRML	86.91 ± 0.03 ^c^	48.24 ± 0.09 ^c^	40.00 ± 0.11 ^c^	59.89 ± 0.08 ^c^
CREL	106.90 ± 0.06 ^d^	45.80 ± 0.07 ^d^	19.15 ± 0.45 ^d^	16.00 ± 0.09 ^d^
CNMF	196.56 ± 0.12 ^e^	23.69 ± 0.19 ^e^	57.99 ± 0.13 ^e^	10.44 ± 0.06 ^e^
CNEF	577.90 ± 0.10 ^f^	24.07 ± 0.09 ^f^	48.02 ± 0.20 ^f^	29.81 ± 0.01 ^f^
CRMF	74.62 ± 0.33 ^g^	13.96 ± 0.11 ^g^	31.86 ± 0.11 ^g^	53.49 ± 0.34 ^g^
CREF	171.78 ± 0.13 ^h^	41.23 ± 0.08 ^h^	102.88 ± 0.15 ^h^	84.81 ± 0.15 ^h^
IND	81.35 ± 0.23	7.23 ± 0.02	4.34 ± 0.05	3.82 ± 0.09

CNML—methanol–acetone–water (3:1:1, *v*/*v*) extract of the leaves of *C. nebrodensis*; CNEL—60% ethanol extract of the leaves of *C. nebrodensis*; CRML—methanol–acetone–water (3:1:1, *v*/*v*) extract of the leaves of *C. roseus*; CREL—60% ethanol extract of the leaves of *C. roseus*; CNMF—methanol–acetone–water (3:1:1, *v*/*v*) extract of the fruits of *C. nebrodensis*; CNEF—60% ethanol extract of the fruits of *C. nebrodensis*; CRMF—methanol–acetone–water (3:1:1, *v*/*v*) extract of the fruits of *C. roseus*; CREF—60% ethanol extract of the fruits of *C. roseus*; IND—Indomethacin. Values were presented as mean ± standard deviation (n = 3). The letters indicate whether a significant difference among the samples existed. The same letters mean no significant difference, and different letters mean significant difference. One-Way ANOVA tests, followed by Tukey’s multiple comparison test, *p* < 0.05.

**Table 6 molecules-27-02907-t006:** Minimal inhibitory concentration (MIC [μg/mL]) values of the extracts obtained from the fruits or leaves of *C. roseus* or *C. nebrodensis*. Bold indicates the better antibacterial activity of *C. roseus* or *C. nebrodensis* extracts compared to Gallic acid.

Sample	*S. aureus*	*S. epidermidis*	*E. coli*	*P. granulosum* PCM 2462	*C. acnes* PCM 2334	*C. acnes* PCM 2400
CNML	nt *	nt	nt	nt	nt	nt
CNEL	nt	nt	nt	nt	nt	nt
CRML	>4000	>4000	-	2000	2000	500
CREL	1000	2000	-	1000	2000	4000
CNMF	500	250	-	**500**	**250**	250
CNEF	>4000	2000	-	4000	1000	4000
CRMF	>4000	>4000	-	2000	4000	1000
CREF	>4000	>4000	-	2000	>4000	>4000
Gallic acid	2000	4000	2000	1000	500	1000

CNML—methanol–acetone–water (3:1:1, *v*/*v*) extract of the leaves of *C. nebrodensis*; CNEL—60% ethanol extract of the leaves of *C. nebrodensis*; CRML—methanol–acetone–water (3:1:1, *v*/*v*) extract of the leaves of *C. roseus*; CREL—60% ethanol extract of the leaves of *C. roseus*; CNMF—methanol–acetone–water (3:1:1, *v*/*v*) extract of the fruits of *C. nebrodensis*; CNEF—60% ethanol extract of the fruits of *C. nebrodensis*; CRMF—methanol–acetone–water (3:1:1, *v*/*v*) extract of the fruits of *C. roseus*; CREF—60% ethanol extract of the fruits of *C. roseus;* nt * not tested due to inactivity in the diffusion test.

**Table 7 molecules-27-02907-t007:** The cytotoxic effect of the extracts obtained from the leaves and fruits of *C. nebrodensis* (CNML, CNEL and CNMF, CNEF) and *C. roseus* (CRML, CREL and CRMF, CREF) (CC_50_) and calculated therapeutic indexes (TIs). The CC_50_ value were expressed as mean values ± SD from three separate experiments.

Extract	CC_50_ (μg/mL) ^a^	*S. aureus* ATCC 25923	*E. epidermidis* ATCC 12228	*E. coli* ATCC 25992	*P. granulosum* PCM 2462	*C. acnes* PCM 2334	*C. acnes* PCM 2400
		Therapeutic Indexes (TIs) ^b^					
CNML	~125	nd ^c^	nd ^c^	nd ^c^	nd ^c^	nd ^c^	nd ^c^
CNEL	~1000	nd ^c^	nd ^c^	nd ^c^	nd ^c^	nd ^c^	nd ^c^
CRML	>1000	~0.25	~0.25	nd ^c^	~0.5	~0.5	~2
CREL	~1000	~1	~0.5	nd ^c^	~1	~0.5	~0.25
CNMF	300.20 ± 1.93	~0.6	~1.2	nd ^c^	~0.6	~1.2	~1.2
CNEF	~1000	~0.25	~1	nd ^c^	~0.25	~1	~0.25
CRMF	347.90 ± 2.21	~0.09	~0.09	nd ^c^	~0.17	~0.09	~0.34
CREF	>1000	~0.25	~0.25	nd ^c^	~0.5	~0.25	~0.25

^a^ CC_50_: the concentration that leads to a 50% reduction of cell viability; ^b^ TIs: the ratio between CC_50_ and MIC values; ^c^ nd: not determined due to lack of antibacterial activity.

**Table 8 molecules-27-02907-t008:** Belonging to clusters (CLF) n = 8 of phytochemicals content.

Clusters (CLF)	TPC Mean	TPC SD	TPAC Mean	TPAC SD	TFC Mean	TFC SD
CLF1	125.44	9.91	58.60	1.68	36.82	20.91
CLF2	64.99	17.66	16.91	6.18	21.67	13.13
CLF3	14.98	6.40	2.98	0.98	2.72	1.15
Total	67.60	43.64	23.85	22.67	20.72	17.43

TPC—Total phenolic content; TPCA—Total phenolic acids content; TFC—Total flavonoid content.

**Table 9 molecules-27-02907-t009:** Belonging to clusters (CLF) n = 8 of antioxidant activity tests.

Clusters (CLF)	DPPH Mean	DPPH SD	ABTS Mean	ABTS SD	CHEL Mean	CHEL SD
CLF1	27.53	6.49	15.96	7.18	33.98	6.16
CLF2	44.65	8.28	40.60	9.98	82.74	24.82
CLF3	121.71	5.54	110.59	14.81	200.53	6.39
Total	59.63	39.54	51.94	38.83	99.99	67.66

CHEL—Metal chelating activity.

**Table 10 molecules-27-02907-t010:** Belonging to clusters (CLF) n = 8 of enzyme inhibitory activity tests.

Clusters (CLF)	LPO Mean	LPO SD	HYAL Mean	HYAL SD	COX-1 Mean	COX-1 SD	COX-2 Mean	COX-2 SD
CLF1	80.77	8.69	31.10	24.24	35.93	5.76	56.69	4.53
CLF2	172.31	46.92	37.32	9.57	60.82	34.25	40.57	34.53
CLF3	451.43	178.86	41.14	24.13	31.17	23.84	26.75	4.33
Total	219.20	166.33	36.72	14.87	47.18	28.36	41.14	25.40

LPO—Lipoxygenase inhibition; HYAL—Hyaluronidase inhibition.

## Data Availability

Data available on request.

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
