# Peer review of "Can Extracts from the Leaves and Fruits of the Cotoneaster Species Be Considered Promising Anti-Acne Agents?"

_molecules, 2022, doi:10.3390/molecules27092907_

Round 1

Reviewer 1 Report

The authors have made the necessary corrections. However, some clarifications and amendments are required.

Since the authors analysed data using one-way ANOVA and Tukey test, shouldn't the authors be able to identify which one was significantly different from which by using just a, ab, and c...for all (Tables 1 and 4)? Is it common to describe each (from a to h)?

Moderate English editing is still needed as there were some mistakes in the text. For example, L160, "Table 1 showed" should be "Table 1 shows" and L169, L178, L237, L312, L313, etc. 

Reviewer 2 Report

The present manuscript was a revised version of the previously submitted article. I had gone through the revised manuscript and author had provided some rational responses towards my previous queries. Although I was not fully satisfied with the answers provided by authors, this manuscript could be recommended to accept for publication in Molecules. Since there were no in vivo data and clear mechanism study, I would suggest this study to be accepted as a note. Alternatively, authors have to address these points clearly in the text.

Reviewer 3 Report

This is a revised manuscript. It describes the LC-MS analysis of two plants' natural products and their biological activities including antioxidant, enzyme inhibitory, antibacterial, and cytotoxic activities. Based on these experimental results, it is not easy to connect the reported activities to anti-acne function. Therefore, a curious but cautious conclusion is also preferred. 

Author Response

This manuscript is a resubmission of an earlier submission. The following is a list of the peer review reports and author responses from that submission.

Round 1

Reviewer 1 Report

The manuscript determined the chemical compositions and biological activities of the leave and fruit extracts isolated from Cotoneaster species. Overall, the study is comprehensive, and the work appears to have been performed in an acceptable manner. However, it will be useful to the readers if the authors could consider the following points.

Minor English editing is needed as there were some mistakes in the text. For example, P3, L152: “also used in in the...” or P5, L213: “This results is”. It should be either “These results are” or “This result is”. Please check it throughout the manuscript.

The introduction might be too lengthy. Some paragraphs, however, is too short, e.g. P2, L52-54. Suggest removing some less important information or combining similar contents in the same paragraph.

P1, L37-38 and L43: Please provide the full name for the first mentioned.

P1, L42, P3 L115: “As reported” change to “As reported by”

P2, L49: Any references to support such a claim?

P5, L200: “dw”. Dry weight?

P5, L203: “were lowest (xxx) than” change to “were lower (xxx) than”.

P6, L255: Any references? Or it was from Kiecel and co-authors [23]? If yes, please insert this reference after the sentence.

P10, L314: Not sure I follow this. What were the “identified polyphenols” here referred to, and where was 10 mg/g? Also, “in this extract”. Please be specific, 60% ethanol?

P10, L316-318: Suggest inserting value after each compound.

P10, L318 & L325: Delete “lowest”.

P12, L353-355: This is a nice summary showing the compounds having anti-acne properties. Perhaps, the authors could consider rephrasing it to “Among the identified compounds found in C. nebrodensis and C. roseus, chlorogenic acid......and quercetin [54] have been reported to show anti-acne activity”.

P12, L357-400: It is good to describe the purpose of this experiment briefly. However, it seems too long for a justification, or some seem like a discussion. The authors perhaps could reduce the text to be more concise and present the results before discussing the findings. This applies to other sections.

P22, L742 and Fig 7: Please label CLF1-CLF3 in Fig. 7.

P23, L744: Is the hierarchical cluster analysis stated in the methodology?

P24, L769-770: Please rephrase.

P28, L845: “Total phenolic acid (TPAC) content” change to “Total phenolic acid content (TPAC)

P29, L930 & L931: “24 h in 37°C” change to “at 37°C for 24 h” or “at 37°C for 48 h”

Table 1: Any significantly differences among samples for total phenolic content, etc.? Has this been statistically analyzed?

Tables 8, 9 and 10: Please cite them in the text.

Table 10: No label for CLF?

Figure 8: Is this needed since the content has been presented in Tables 8, 9 and 10? Should be it in the supplementary material?

Conclusion: It seems like an introduction with some discussion. The conclusion should highlight the main findings/conclusion of this study and future work or potential applications.

Reviewer 2 Report

This manuscript was aimed to investigate the chemical composition as well as biological activities, including the antioxidant, antibacterial, enzyme inhibitory and cytotoxic activities of leaves and fruits extracts of C. roseus and C. nebrodensis. The results indicated that these extracts were rich in phenolic compounds and biologically active. Authors performed many experiments and these experimental data were really interesting, however, the present results only displayed preliminary advances and no significant improvements on the bioactive components discovery related fields could be observed. In summary, this manuscript is not recommended to accept for publication in Molecules. In addition, there are several major comments to be addressed as following.

  1. The language of this manuscript was well, however, there were still some minor problems to be observed in the text. For examples, the Introduction and Conclusion sections were too lengthy. Authors have to combine some paragraphs into one and rewrite these in a more concise manner. Some minor mistakes could also be observed in the References section, such as ref 3. Authors have to check and revise these errors carefully.
  2. The LC-MS quantification data was not so reliable since many compounds were not quantified with the same standards. The amounts in Tables 2 and 3 should be illustrated as “tentatively determined”.
  3. Although various antioxidant assays were performed for different extracts, there was not any single compound to be studied in this research. The present data could not be the bases for the further development of any new lead drugs.
  4. In the anti-bacterial activity examination section, there were not any data for the positive controls to be observed. In addition, these anti-bacterial data were not significant.
  5. The experimental details of HCA analysis were not included in the Experimental section.
  6. There were not any in vivo data in this manuscript. In addition, the reasonable concentrations of these compounds were difficult to be reached in in vivo model. Authors should discuss this problem if they wish to resubmit this manuscript.

Reviewer 3 Report

  • Introduction is too long - contains a lot of unnecessary information. Needs to be condensed down and written with a clear direction.
  • Materials and methods should be before results and discussion.
  • Extensive editing of English language and style required. Difficult to read the content with so many grammatical errors. I have edited the first paragraph (see attached), use this as an example for the rest of the manuscript.
